# UPS: Efficiently Building Foundation Models for PDE Solving via Cross-Modal Adaptation

**Junhong Shen**                                                                          *junhongs@andrew.cmu.edu*
*Machine Learning Department, Carnegie Mellon University*

**Tanya Marwah**                                                                          *tmarwah@andrew.cmu.edu*
*Machine Learning Department, Carnegie Mellon University*

**Ameet Talwalkar**                                                                          *talwalkar@cmu.edu*
*Machine Learning Department, Carnegie Mellon University*

**Reviewed on OpenReview:** *https://openreview.net/forum?id=0r9mhjRv1E*

## Abstract

We present Unified PDE Solvers (UPS), a data- and compute-efficient approach to developing unified neural operators for diverse families of spatiotemporal PDEs from various domains, dimensions, and resolutions. UPS embeds different PDEs into a shared representation space and processes them using a FNO-transformer architecture. Rather than training the network from scratch, which is data-demanding and computationally expensive, we warm-start the transformer from pretrained LLMs and perform explicit alignment to reduce the modality gap while improving data and compute efficiency. The cross-modal UPS achieves state-of-the-art results on a wide range of 1D and 2D PDE families from PDEBench, outperforming existing unified models using 4 times less data and 26 times less compute. Meanwhile, it is capable of few-shot transfer to unseen PDE families and coefficients.

## 1 Introduction

Partial Differential Equations (PDEs) play a pivotal role in modeling and understanding real-world phenomena, such as fluid dynamics and heat transfer. Although there exists a rich body of classical PDE solvers (Boyd, 2001; LeVeque, 2007; Moukalled et al., 2016) that are effective and mathematically proven, these solvers often incur substantial computational costs when used in practice, as they need to be re-run every time a coefficient or boundary condition changes. This motivates the development of *neural operators* (Li et al., 2020a; Chen & Chen, 1995; Lu et al., 2019), which use neural networks to approximate a solution map for a PDE family and can generalize to different initial/boundary conditions or coefficients. While existing neural operators (Lippe et al., 2023; Hao et al., 2023a; Marwah et al., 2023) have demonstrated strong performance on various practical benchmarks (Takamoto et al., 2022; Gupta & Brandstetter, 2022), most of them are designed to work with a *single PDE family*. Training a separate model for each PDE family remains costly.

Several recent works, such as Subramanian et al. (2023), MPP (McCabe et al., 2023), and DPOT[1] (Hao et al., 2024), have taken initial steps towards developing foundation models for PDE solving, learning unified operators that transfer across PDE families. These models are *pretrained from scratch* using extensive amounts of data and compute. For example, MPP is trained with over 80,000 PDE trajectories on 8 NVIDIA H100 GPUs for 200,000 steps. Despite the development costs, the resulting models are limited in generalization ability—all existing unified models focus on pretraining with 2D PDEs. Finally, as these

---

[1]This work was done at the same time as ours.

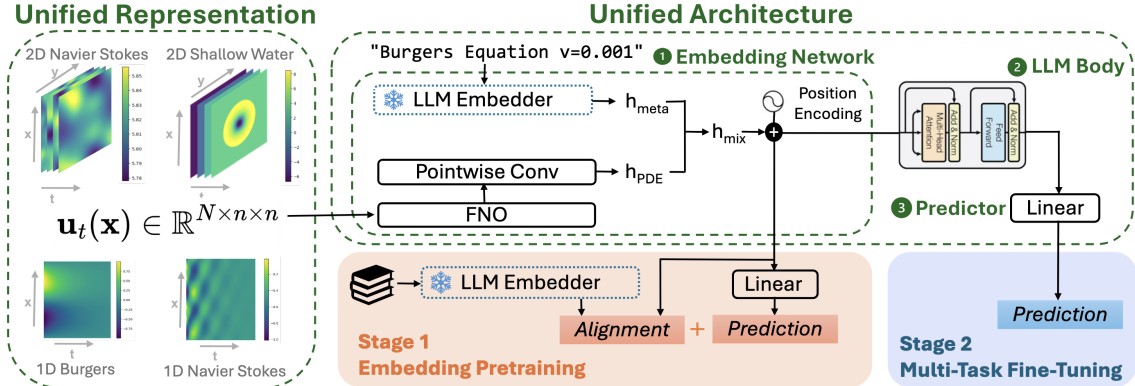

Figure 1: To adapt pretrained LLMs for PDE solving, UPS first transforms PDE of different dimensions, channels, and resolutions into a *unified representation* (left panel). Then, the data is processed with a *unified architecture* that integrates FNO layers, PDE metadata, and LLMs (right panel). The architecture is trained in two stages. In stage 1, we pretrain the embedding network using a joint loss that simultaneously optimizes (i) the distribution similarity between PDE features and text embeddings to align the modalities, and (ii) the prediction performance of extracted PDE features. In stage 2, we fine-tune the entire model on a dataset that combines multiple families of spatiotemporal PDEs with varying domain dimensions, initial/boundary conditions, and coefficients. UPS achieves competitive results with significantly better sample-efficiency than existing methods.

models are developed using only PDE trajectories, they do not leverage meta information that could help distinguish between various PDE families, such as the name of the family and the coefficients.

We present Unified PDE Solvers (UPS), which learns unified neural operators for complex time-dependent PDEs with improved efficiency and generalization ability. Unlike existing efforts that train models from scratch, we propose a novel method to adapt pretrained Large Language Models (LLMs) to PDE solving. This is inspired by the line of work that repurposes LLMs for scientific domains like mathematics (Lewkowycz et al., 2022), computational biology (Shen et al., 2023; Vinod et al., 2023; Joachimiak et al., 2023), and chemistry (Bran et al., 2023; Shen et al., 2024). These works not only show how LLMs utilize both text and non-text information to solve scientific problems and transfer effectively to unseen tasks, but also provide strong evidence that the general-purpose pretraining and inductive biases of LLMs could substantially reduce the sample complexity needed for adaptation.

Concretely, UPS adapts pretrained LLMs to time-evolution operators that map the current state of a PDE to its future state for general spatiotemporal PDEs (see Section 3 equation 1 for definition) using two key designs (see also Figure 1 and Section 3):

1. We propose a **unified data representation scheme** to align PDEs with varying dimensions and physical quantities into the same feature space. Given the space and time discretization $\boldsymbol{u} = \{\boldsymbol{u}_t(\boldsymbol{x})\}_{t=0}^T$, where $\boldsymbol{x} \in \mathbb{R}^d$ is the spatial variable and $\boldsymbol{u}_t(\boldsymbol{x})$ is the state variable, UPS homogenizes $\boldsymbol{u}_t(\boldsymbol{x})$ from diverse PDEs into a shared "superspace" $\mathbb{R}^{N \times n^{d_{\max}}}$, where $d_{\max}$ is the maximum dimension of $\boldsymbol{x}$ among all PDEs considered, $N$ is the superset of the physical quantities, and $n$ is the resolution. This framework of embedding lower-dimensional PDEs in a higher dimension enables UPS to model cross-dimensional PDEs simultaneously and distinguishes us from all existing unified operators, which do not consider low dimensional PDEs in 1D.

2. We employ a **unified network architecture** to predict $\boldsymbol{u}_{t+1}(\boldsymbol{x})$ based on $\boldsymbol{u}_t(\boldsymbol{x})$. To leverage pretrained LLMs, we design a three-way architecture that consists of (i) a FNO (Li et al., 2020a) based embedding network to convert PDE data into resolution-invariant, sequential features; (ii) an LLM body to process the PDE features and the text embeddings of the PDE metadata; and (iii) a prediction network to generate the final output. Inspired by previous cross-modality adaptation work (Shen et al., 2023), we employ a two-stage align-then-refine process for model training. However, we improve the align stage by using a joint loss that adds feature extraction on top of alignment to pretrain the embedding network. We improve the refine stage by fine-tuning on a collection of PDE tasks rather than a single task. Our enhanced

workflow outperforms naive transfer and previous cross-modal approaches, reducing both the data and compute needed for training.

By design, UPS can handle diverse PDE families, data dimensions, channels, and resolutions. More crucially, by warm-starting with pretrained LLM weights and applying explicit alignment, UPS strikes a balance between effectiveness and efficiency—it achieves state-of-the-art performance across 9 datasets from PDEBench (Takamoto et al., 2022) (7 in-distribution, 2 out-of-distribution), using about 20,000 training trajectories, a single A6000, 60,000 train steps, and under 100 GPU hours. This means that we achieve better results than existing unified models using 4 times less data and 26 times less compute.

Beyond prediction accuracy, we confirm that UPS preserves key properties of neural operators, such as grid- and resolution-invariance. We also show that UPS is compatible with a variety of LLM backbones, including RoBERTa (Liu et al., 2019), T5 (Raffel et al., 2020), and CLIP (Radford et al., 2021), and demonstrates better performance when scaled to larger backbones. We believe that the model-agnostic design of UPS offers a systematic approach to harnessing the advancements in LLMs for PDE solving, and it takes a further step towards building generalized foundation models for more complex physical systems efficiently. Code is available at https://github.com/sjunhongshen/UnifiedPDESolvers.

## 2 Related Work

Recent years has seen a variety of neural-network-based methods for approximating PDE solutions. Hybrid solvers (Hsieh et al., 2019; Bar-Sinai et al., 2019; Kochkov et al., 2021) apply classical solvers like finite element/volumn methods (LeVeque, 2007; Moukalled et al., 2016) to a low-resolution grid and use neural networks to predict the correction terms. Others directly approximate the PDE solutions with neural networks (Sirignano, 2017; Raissi et al., 2019; Khoo et al., 2021; Shen et al., 2022), using variational losses (Yu et al., 2018) or physical constraints defined by the PDE (Raissi et al., 2019; Bruna et al., 2024). Being mostly equation-specific, these methods can solve one PDE at a time. The learned models do not apply to other PDEs in the same family, let alone other families.

A more general approach involves learning neural operators (Lu et al., 2019; Li et al., 2020a;b) which approximate an infinite-dimensional operator between two functional spaces. For time-dependent PDEs, a neural operator maps the current state of a PDE to the next state, with quantities like initial conditions provided as input. Neural operators can be implemented using any architecture. For example, Fourier neural operator (FNO) (Li et al., 2020a) uses convolution-based integral kernels evaluated in the Fourier space. Other works also use transformer models (Cao, 2021; Li et al., 2022; Hao et al., 2023a) or U-Net (Lippe et al., 2023). Learning neural operators enables solving an entire family of PDE and they can easily adapt to new parameterizations of a PDE without fine-tuning. However, the learned operators cannot extend to different PDE families.

To facilitate operator transfer across PDE families, two recent works develop large pretrained models for multiple physical systems: Subramanian et al. (2023) train FNOs on steady-state linear PDEs with periodic boundary conditions; McCabe et al. (2023) design a new transformer architecture based on the axial attention (Ho et al., 2020) and train it using various 2D non-linear, time-dependent PDEs. While these methods show that a unified operator can outperform single-family operators, they are limited in two aspects. First, existing unified methods consider mainly 2D PDEs for pretraining and evaluation. In contrast, UPS leverages a unified representation scheme to tackle both 1D and 2D PDEs. This method can be also extended to any $d$-dimensional systems in theory. Second, existing methods pretrain large models from scratch and necessitate extensive GPU resources and pretraining data, which can be prohibitive to collect for high-dimensional complex PDEs. However, by adapting from pretrained LLMs and closing the modality gap between text and PDE efficiently, UPS achieves competitive results using 4x less data and 26x less compute.

Beyond the aforementioned works, DPOT (Hao et al., 2024) was developed concurrently with our work and presents an auto-regressive denoising strategy for pretraining. While DPOT has shown better transferability to unseen PDE tasks than MPP, it shares the same limitations of focusing on 2D problems for pretraining and requiring large amount of data and compute (8 A800 GPUs for 500,000 steps).

A final work that is related to ours is ORCA (Shen et al., 2023), which proposes a general workflow for adapting pretrained language/vision transformers to non-text/vision inputs. While ORCA uses PDEBench in its evaluation, it is not tailored to PDE solving and requires adapting a separate model for every dataset. The resulting models are not grid- or resolution-invariant, which are key properties of neural operators and achieved by UPS. Moreover, by learning from multiple PDEs and sharing knowledge across families, UPS obtains significantly better empirical results than ORCA.

## 3  Methodology

Our goal is to train unified neural operators for spatiotemporal PDEs with varying domain dimensions, coefficients, initial and boundary conditions. These PDEs could model a range of quantities that evolve over time, from scalars (e.g., pressure, density) to vectors (e.g., velocity). To achieve this, we propose UPS, which consists of a unified way to represent the PDE and a LLM-based network to model them.

### 3.1  Unified Data Representation

We model PDEs that follow the general form:

$$\frac{\partial \boldsymbol{u}(t, \boldsymbol{x})}{\partial t} = L\left(\boldsymbol{u}(t, \boldsymbol{x}), \frac{\partial \boldsymbol{u}(t, \boldsymbol{x})}{\partial \boldsymbol{x}}, \frac{\partial \boldsymbol{u}(t, \boldsymbol{x})}{\partial \boldsymbol{x}^2}, \cdots\right)$$
$$u(0, \boldsymbol{x}) = u_0(\boldsymbol{x}) \quad B\left(\boldsymbol{u}(t, \boldsymbol{y})\right) = h(y) \tag{1}$$

where $\boldsymbol{x} \in \Omega \subset \mathbb{R}^d$ is the spatial variable, $\boldsymbol{u} : [0, T] \times \Omega \to \mathbb{R}^{d_u}$ is a time-varying function defined over the domain $\Omega$ for finite time $T$. Here, $L$ is a (possibly non-linear) operator which acts on $\boldsymbol{u}$ and multiple partial derivatives of $\boldsymbol{u}$ w.r.t the spatial variable $\boldsymbol{x}$. $\boldsymbol{u}_0(\boldsymbol{x}) : \Omega \to \mathbb{R}^{d_u}$ denotes PDE's initial condition, and the operator $B$ defines the boundary condition where $\boldsymbol{y} \in \partial\Omega$ is a point on domain's boundary, and $h : \partial\Omega \to \mathbb{R}^{d_u}$ defines the given function over the boundary[2]. PDE families in this form include Navier-Stokes equations, Reaction-Diffusion equations, Burgers equations, and many others that describe phenomena like fluid dynamics and heat flow over time. They also constitute most PDE benchmarks in the field of machine learning (Takamoto et al., 2022; Tu et al., 2022; Roberts et al., 2021).

Consider a set of $S$ spatiotemporal PDEs $\{\boldsymbol{u}^s\}_{s=1}^S$. Here, each $\boldsymbol{u}^s = \{\boldsymbol{u}_t^s(\boldsymbol{x})\}_{t=1}^{T_s}$ is a solution to a PDE of the form defined in equation 1 such that for all $t \in [T_s]$, we have $\boldsymbol{u}_t^s(\boldsymbol{x}) \in \mathbb{R}^{d_u^s}$ and $\boldsymbol{x} \in \Omega^s \subset \mathbb{R}^{d^s}$, where $d^s$ is the dimension of the PDE $s$. For each $\boldsymbol{u}_t^s$, we assume that we have an $n$-point discretization of the functions $\{\boldsymbol{u}_t^s\}_{t=1}^{T_s}$ at points $W_n^s = \{\boldsymbol{x}_1^s, \boldsymbol{x}_2^s, \cdots, \boldsymbol{x}_{n^{d^s}}^s\}$, where each $\boldsymbol{x}_i^s \in \mathbb{R}^{d^s}$. That is, for each PDE $s \in S$ and $t \in T_s$, we have the realization of the function $\boldsymbol{u}_t^s$ on a grid with each dimension divided into $n$ parts, thus giving rise to $n^{d^s}$ points in the set. We assume that $n$ is constant across PDE families. We note that this value $n$ is a hyperparameter, and ideally should be the minimum number of points that work well for *all the PDEs* considered, for example, low-viscosity Navier-Stokes may require more discretization points compared to the high viscosity counterparts. Denote the set of $N$ physical quantities considered for each PDE as $V = \{v_1, v_2, \cdots v_N\}$. Our goal is to learn an operator $\mathcal{G}_\theta$ which, for a given PDE $s$, predicts the state of the PDE at time $t + 1$ based on its state at time $t \in [T_s]$, i.e., $\hat{\boldsymbol{u}}_{t+1}^s(\boldsymbol{x}) = \mathcal{G}_\theta(\boldsymbol{u}_t^s(\boldsymbol{x}))$. We thus need a unified representation for the inputs so a model can handle different quantities at once.

**Unifying Dimension**  Let $d^s$ denote the dimension of the PDE $s$ and $d = \max_{s \in S} d^s$. We want to represent all datasets in $\mathbb{R}^d$. Thus, for PDEs with $d^s < d$, the final $d - d^s$ coordinates of $\boldsymbol{x}_i^s \in W_n^s$ are set to zero. In this work, we mainly consider PDEs defined over one- and two-dimensional domains, i.e., $d^s \in \{1, 2\} \ \forall s \in S$. Hence, for PDEs with $d^s = 1$, the point $\boldsymbol{x} \in \Omega^s$ is represented with the 2D-coordinate $(x, 0)$. Note that our methodology to unify the total number of dimensions in the PDE is general and can be adapted to PDEs defined in higher-dimensional domains as well. In the following, we will denote $\boldsymbol{u}_t^s(\boldsymbol{x})$ as the value of the function $\boldsymbol{u}_t^s$ on all the points in $W_n^s$, unless stated otherwise.

**Unifying Physical Quantities**  We consider a fixed set $V = \{v_1, v_2, \cdots v_N\}$ of $N$ physical quantities and train our model on the quantities that belong to $V$ for each PDE. The quantities we consider in this paper are velocity (in both $x$ and $y$ directions), pressure, and density, and they are the superset of all quantities for

---

[2]Unless stated otherwise, we assume that the value of the function is 0 at the boundary, i.e, $h(y) = 0$ for all $y \in \partial\Omega$.

the PDE families we evaluate. This leads to $N = 4$. If a dataset does not use a particular quantity, the entire dimension corresponding to it is set to 0.

With the above procedure, we lift every PDE to a unified space so $\boldsymbol{u}^s \in \mathbb{R}^{T_s \times N \times n^d} \ \forall s \in S$. To obtain the datasets for forward prediction, we generate input-output pairs via autoregressive teacher-forcing: for each time step $t \in [T_s]$, we use $\boldsymbol{u}_t^s$ to predict $\hat{\boldsymbol{u}}_{t+1}^s$, yielding $T_s - 1$ pairs of data from a single trajectory. We append the coordinates of each $\boldsymbol{x}_i^s \in W_n^s$ to the input and maintain an output mask to mask out the zero-padded dimensions when computing the loss.

## 3.2 Unified Architecture

Transformer models have demonstrated success in various domains like natural language (e.g., Touvron et al., 2023), vision (e.g., Dosovitskiy et al., 2021), and audio processing (e.g., Lu et al., 2023). In this work, we explore the potential of transformers for PDE solving. We break down the UPS architecture into 3 parts: an embedding network that transforms the unified representation into sequence features; the model body, consisting of the pretrained LLM layers; and a predictor that generates the prediction (Figure 1).

**FNO Embedding Network** The embedding network plays two roles. First, it projects the PDE $\boldsymbol{u}_t^s(\boldsymbol{x})$ into the LLM's sequential embedding space $\mathbb{R}^{l \times e}$, where $l$ denotes the sequence length of the embedded features and $e$ denotes the LLM's hidden dimension. Second, it should extract key features of the PDE input to enable subsequent transformer layers to make predictions. Therefore, we design a PDE-specific embedding network with FNO layers for feature extraction, a linear layer for dimensionality matching, and a concatenation operator for adding metadata (Figure 1).

We use FNO due to its strong empirical performance (Li et al., 2020a; Takamoto et al., 2022) and its ability to extract resolution-invariant features. As we consider maximum two-dimensional PDEs in this work, we use a series of 2D FNO layers with $l$ channels to obtain PDE features in $\mathbb{R}^{l \times n^d}$. Then, to map the FNO output to the LLM's embedding dimension, we apply a pointwise convolution with input channel $n^d$, output channel $e$, kernel size 1, stride 1. This yields the desired sequential features $h_{\text{PDE}} \in \mathbb{R}^{l \times e}$.

Since UPS is intended to handle diverse data from various generating sources, we leverage the PDE's metadata in addition to the input dynamics. The motivation is that LLMs can use the textual information to better understand the context and characteristics of different PDEs. To implement this, we specify the metadata in the form "`[PDE family] [coefficients]`" which is embedded into sequential features $h_{\text{meta}}$ using the LLM's tokenizer and text embedding layer. We then concatenate the meta features and the PDE features to get $h_{\text{mix}} := [h_{\text{meta}}, h_{\text{PDE}}]$. Finally, we apply positional encoding and layer norm to $h_{\text{mix}}$. This will be the input to the subsequent transformer layers.

In Section 5.3, we perform various ablation studies on the embedding network. We investigate the effect different hyperparameters, such as the channel dimension $l$ in FNO, and show that incorporating metadata improves both prediction performance and generalization ability of UPS.

**Utilizing Pretrained LLMs** The main body of a UPS model consists of pretrained transformer layers from an LLM. Thus, we pass $h_{\text{mix}}$ to the pretrained transformer layers, which produce the hidden states $\hat{h} \in \mathbb{R}^{l \times e}$. Since there is no causal structure in the spatial dimensions of a PDE, we do not apply autoregressive masking to $h_{\text{mix}}$ and allow the embedded features to attend to each other.

Our design provides flexibility for using different LLMs as the model body. We show experiment results with multiple LLMs in Section 5.3. While different LLMs have different performance, they are competitive with existing baselines. We also show that adapting from pretrained weights outperforms training the same architecture from scratch, so UPS is especially useful for low-data regimes.

**Linear Predictor** Finally, we define a prediction head to transform the hidden state of the LLM body $\hat{h}$ to the predicted next step of the input $\hat{\boldsymbol{u}}_{t+1}^s(\boldsymbol{x}) \in \mathbb{R}^{N \times n^d}$ (we predict all the physical quantities in the set $V$). This is achieved by averaging over the sequence dimension of $\hat{h}$ to get shape $\mathbb{R}^e$, applying a linear layer to map it to $\mathbb{R}^{Nn^d}$, and reshaping the results to obtain the final prediction $\hat{\boldsymbol{u}}_{t+1}^s(\boldsymbol{x})$. The linear predictor is shared for all PDEs.

## 4   Full Workflow and Training

We train UPS in two stages. In the first stage, we train the embedding network to align $h_{\mathrm{mix}}$ with the LLM's embedding space. This is because LLMs are trained for the text modality, which has distinct characteristics and features from physical processes like fluid dynamics and heat flow. Stage 1 reduces the modality gap to prevent the distortion of pretrained weights. Next, we fine-tune the entire model on a dataset of multiple families of spatiotemporal PDEs.

**Stage 1: Embedding Pretraining**   Intuitively, there is a modality gap between text data used to train general-purpose LLMs and PDEs. Previous work has also shown that directly fine-tuning pretrained LLMs on non-text inputs can result in suboptimal performance (Lu et al., 2022). To address this, Shen et al. (2023) introduced ORCA, which performs distribution matching before fine-tuning to enable cross-modal adaptation. That is, given a randomly initialized embedding network, we first pretrain it to minimize the distribution distance between the embedding network's output—in our case $h_{\mathrm{mix}}$—and the text embeddings of a external reference NLP dataset, which we denote as $h_{\mathrm{LM}}$. This process makes the cross-modal distribution resemble the text distribution that the LLM is pretrained on. Following the ORCA work, we use the CoNLL-2003 dataset (Sang & Meulder, 2003) as the reference dataset for alignment.

We propose several PDE-specific improvements to the alignment process. First, unlike ORCA which uses an optimal transport (OT) based metric for measuring the distribution distance, we use the maximum mean discrepancy (MMD) distance for UPS. This is because the OT-based metric requires discrete class labels to compute, making it unsuitable for PDEs. In contrast, MMD acts directly on the features $h_{\mathrm{mix}}$ and is more computationally efficient. Thus, we define

$$\mathcal{L}_{\mathrm{align}} = \|\mu_{\mathcal{D}_{h_{\mathrm{mix}}}} - \mu_{\mathcal{D}_{h_{\mathrm{LM}}}}\|_{L_2} = \mathbb{E}_{\mathcal{D}_{h_{\mathrm{mix}}}}[k(a, a')] - 2\mathbb{E}_{\mathcal{D}_{h_{\mathrm{mix}}}, \mathcal{D}_{h_{\mathrm{LM}}}}[k(a, b)] - \mathbb{E}_{\mathcal{D}_{h_{\mathrm{LM}}}}[k(b, b')] \tag{2}$$

where $k(a, a') = \exp(\|a - a'\|_2 / 2)$ denotes the Gaussian kernel; $\mathcal{D}_{h_{\mathrm{mix}}}$ and $\mathcal{D}_{h_{\mathrm{LM}}}$ denote the distributions of the PDE embeddings $h_{\mathrm{mix}}$ and the reference text embeddings $h_{\mathrm{LM}}$.

Second, to improve the feature extraction ability of the embedding network in the context of our downstream task, we introduce a *task loss* for PDE forward prediction, i.e., the normalized root mean squared (nRMSE) loss between the prediction $\hat{\boldsymbol{u}}_{t+1}^s(\boldsymbol{x})$ and the ground truth $\boldsymbol{u}_{t+1}^s(\boldsymbol{x})$:

$$\mathcal{L}_{\mathrm{task}} = \frac{1}{S} \sum_{s=0}^{S} \frac{1}{T_s} \sum_{t=0}^{T_s-1} \frac{\|\boldsymbol{u}_{t+1}^s(\boldsymbol{x}) - \hat{\boldsymbol{u}}_{t+1}^s(\boldsymbol{x})\|_2}{\|\boldsymbol{u}_{t+1}^s(\boldsymbol{x})\|_2} \tag{3}$$

Thus, the final objective for pretraining the embedding network is the joint loss $\mathcal{L}_{\mathrm{emb}} = \mathcal{L}_{\mathrm{align}} + \mathcal{L}_{\mathrm{task}}$. We show in Section 5.3 that both objectives are essential to the overall performance of UPS.

**Stage 2: Multi-Task Fine-Tuning**   In contrast to most existing neural PDE solvers, which train a separate model for each dataset, UPS is trained using one large dataset consisting of PDE data from multiple generating sources (all of $S$). Hence, after learning the embedding network, we fine-tune the entire model (the embedding network, the LLM body, and the linear predictor) using $\mathcal{L}_{\mathrm{task}}$ defined in equation 3. We evaluate the performance of UPS in Section 5.1 and find it outperforms existing single-dataset neural operators. We also show that UPS generalizes to unseen PDE families and coefficients (Section 5.2)—the zero-shot and few-shot adaptation performance is competitive with models specifically trained on the entire target dataset.

## 5   Experiments

**Data**   We train and evaluate our method using PDEBench (Takamoto et al., 2022). For training, we combine 7 datasets from different PDE families: Burgers Equation (1D), Advection (1D), Diffusion-Sportion (1D), Shallow-Water (2D), compressible Navier-Stokes (1D and 2D), and incompressible Navier-Stokes (2D). We explicitly hold out two families, 1D and 2D Diffusion-Reaction, to evaluate the generalization ability of UPS. The dataset details can be found in Appendix A.1. We autoregressively generate the predictions and use the

Table 1: nRMSEs (lower is better) for in-distribution PDEBench families, with baseline results taken from Takamoto et al. (2022); Shen et al. (2023); McCabe et al. (2023); Hao et al. (2024). '-' means that the result is not available. On all datasets, UPS with RoBERTa-Base (UPS-B) achieves the lowest nRMSEs among all smaller models and UPS with RoBERTa-Large (UPS-L) achieves the lowest nRMSEs among all large models. Numbers are bolded for each model size group.

| | # Params (sorted) | Advection 1D | Burgers 1D | Diffusion-Sorption 1D | Navier-Stokes 1D | Shallow-Water 2D | Navier-Stokes 2D | Incomp Navier-Stokes 2D |
|---|---|---|---|---|---|---|---|---|
| **Single-Family** | | | | | | | | |
| FNO | 466K | 0.011 | 0.042 | 0.0017 | 0.068 | 0.0044 | 0.36 | 0.0942 |
| GNOT | 1.8M | - | - | - | - | 0.0068 | 0.0373 | - |
| OFormer | 1.9M | - | - | - | - | 0.0072 | 0.0521 | - |
| U-Net | 7.7M | 0.67 | 0.34 | 0.15 | 0.72 | 0.083 | 5.1 | 0.1903 |
| ORCA | 125M | 0.0098 | 0.12 | 0.0016 | 0.062 | 0.006 | 0.3549 | 0.1529 |
| **Unified (Small)** | | | | | | | | |
| Unified FNO | 466K | 0.013 | 0.0501 | 0.0041 | 0.0101 | 0.0033 | 0.152 | 0.1064 |
| MPP-B | 116M | - | - | - | - | 0.0024 | 0.0281 | - |
| DPOT-M | 122M | - | - | - | - | 0.0029 | 0.0177 | - |
| UPS-B (**Ours**) | 149M | **0.0027** | **0.0399** | **0.0009** | **0.0056** | **0.0016** | **0.0153** | **0.0931** |
| **Unified (Large)** | | | | | | | | |
| UPS-L (**Ours**) | 387M | **0.0022** | **0.0373** | **0.0009** | **0.0045** | **0.0015** | **0.015** | **0.0924** |
| MPP-L | 409M | - | - | - | - | 0.0022 | 0.0208 | - |
| DPOT-L | 500M | - | - | - | - | 0.0023 | 0.0158 | - |

scale-independent normalized root mean squared error (nRMSE) as the evaluation metric, defined as follows:

$$\text{nRMSE} = \frac{1}{S_{\text{test}}} \sum_{s=1}^{S_{\text{test}}} \frac{\|\boldsymbol{u}^s(\boldsymbol{x}) - \hat{\boldsymbol{u}}^s(\boldsymbol{x})\|_2}{\|\boldsymbol{u}^s(\boldsymbol{x})\|_2} \qquad (4)$$

We preprocess all the PDEs by normalizing each dataset along the channel dimension to ensure the scale of $\boldsymbol{u}_t^s(\boldsymbol{x})$ across datasets is similar[3].

**Baselines** We compare against two sets of baselines: (i) single-family models trained on individual PDE datasets, including the widely used U-Net (Ronneberger et al., 2015), FNO (Li et al., 2020b), the improved version FFNO (Tran et al., 2023), the transformer-based GNOT (Hao et al., 2023b) and OFormer (Li et al., 2023), as well as the cross-modal ORCA (Shen et al., 2023); (ii) unified models trained on multiple datasets, including MPP (McCabe et al., 2023), DPOT (Hao et al., 2024), and a unified FNO trained using data transformed by our unified representation scheme. We note that MPP and DPOT focus on 2D PDEs and are pretrained on 2D Navier-Stokes, Shallow-Water, and Diffusion-Reaction from PDEBench. Subramanian et al. (2023) is not included as a baseline because its models are pretrained on different PDE families (e.g., Poisson's and Helmholtz equations) not present in PDEBench.

**Implementation Details** As noted in Section 3, UPS is compatible with any pretrained LLM. We present our main results using RoBERTa (Liu et al., 2019) and study other backbones in ablation studies (Table 4). We set the embedding FNO channel $l$ to 32. Since the resolution of the 2D datasets in PDEBench is 128, we set the model resolution $n$ to 128 and downsample datasets with higher resolutions. All of our experiments can be run on a single NVIDIA A6000 GPU. See Appendix A.2 for training details. Due to computational constraints, results are based on a single run per network configuration.

## 5.1 Achieving State-of-the-Art Results on PDEBench with Compute Efficiency

We first study the *in-distribution* performance of UPS, i.e., we evaluate UPS on the test splits of the datasets that are used to train UPS, which consists of PDEs that share the same boundary conditions and coefficients with the training samples, but have different initial conditions. The results are shown in Table 1. In general, UPS with RoBERTa ranks first on all 7 datasets and improves the state-of-the-art by an order of magnitude on many 1D datasets. We analyze the results in more details below.

---

[3]We standardize the data by subtracting the mean and dividing by the standard deviation to ensure training stability when using pretrained model weights. For loss computation, we apply an inverse transformation to the outputs to revert them to the original scale. Although data normalization may affect non-linear equations, it is essential to prevent loss explosion during fine-tuning. We leave exploring alternative methods to minimize distortion in non-linear dynamics as future work.

**Compare with Single-Family Operators** We outperform all single-family models like FNO and ORCA, which train a different model for every PDE family. This shows the benefits of learning a versatile neural operator rather than multiple specialized ones, and our model is capable of extracting universal rules when learning to model multiple PDE equations.

**Compare with Unified Operators** We note that existing unified models like MPP and DPOT do not pretrain or evaluate on 1D problems due to the limitation of their data representation. In contrast, UPS embeds low-dimensional PDEs in high-dimensional spaces and model all PDEs uniformly despite the dimensionality difference, achieving state-of-the-art results on all 1D datasets in PDEBench. As for the 2D problems considered, UPS with RoBERTa-Base outperforms MPP-B and DPOT-M, which have similar model sizes. UPS with RoBERTa-Large outperforms MPP-L and DPOT-L. We emphasize that UPS is trained on significantly fewer trajectories per PDE family (<5K) compared to the baselines. Besides, UPS can be run on a single A6000 for less time while maintaining good performance. This shows the data and compute benefits of adapting from pretrained LLMs.

Since MPP and DPOT focus on 2D problems and use a different set of pretraining datasets from ours, we train a 2D-only UPS on all 2D datasets in PDEBench to provide a more direct comparison. The results are shown in Appendix Table 8. Notably, while 2D UPS is still trained with less data (since the other methods use additional datasets like PDEArena (Gupta & Brandstetter, 2022)), our method ranks first on 4 of 8 datasets, outperforming DPOT on 5 of 8 datasets and outperforming MPP on 3 of 4 datasets.

Recall also that we train a 2D unified FNO using the datasets processed by our dimension unification scheme. The unified FNO does not always outperform single-family FNOs, especially on 1D tasks, possibly because the network is 2D, and the relatively simple architecture might not be able to extract shared information across PDE families and leverage it to improve performance. More crucially, UPS outperforms unified FNO on all datasets, showing the efficacy of our LLM-based architecture.

**Scaling Up LLM Backbones** To study the scaling behavior of our method, we adapt from both RoBERTa-Base (149M parameters) and RoBERTa-Large (387M parameters) and report the results in Table 1. The first observation is that the two versions of UPS all outperform baselines of similar sizes, achieving both effectiveness and efficiency. Besides, UPS-Large generally outperforms UPS-Base, which shows that scaling up the backbone has the potential to yield better results.

In addition to prediction errors in Table 1, we visualize some of the UPS outputs in Appendix A.4 and show that it is indeed able to capture the key features and dynamics of different PDE families. For efficiency metrics, we report the training compute requirement, FLOPs, and inference time for UPS in Appendix A.2.2. Compared to existing work, our method has lower FLOPs and shorter inference time. This shows that our method can be deployed in practical environments where both computational efficiency and speed are critical.

## 5.2 Generalizing to Unseen PDEs with Data Efficiency

In this section, we investigate the generalization (*out-of-distribution*) performance of UPS under three scenarios: (i) unseen PDE families, (ii) PDEs belonging to the training families but with different coefficients, and (iii) PDEs with higher-resolution grids. Unless otherwise specified, UPS-B is used.

**Unseen PDE Families** As mentioned earlier, we hold out the Diffusion-Reaction equations from developing UPS. We first directly evaluate UPS on these two tasks and report the zero-shot transfer performance. Then, we study few-shot transfer by randomly sampling $k \in \{10, 100\}$ trajectories from the training sets of the held-out tasks and use them to fine-tune UPS. Lastly, we report the fine-tuning results with the full training dataset. The results are

Table 2: Zero- and few-shot transfer performance of UPS on unseen PDE families and coefficients. Our few-shot results are competitive with baselines trained with more data. UPS-B refers to UPS with RoBERTa-Base.

| # Samples | Model | Unseen PDE Families | | Unseen Coefficients | |
| | | 1D Diff-React | 2D Diff-React | 1D Burgers $\nu = 1.0$ | 2D Navier-Stokes $M = 1, \eta = \zeta = 0.1$ |
|---|---|---|---|---|---|
| 0 | UPS-B | **0.0557** | **1.0593** | **0.0566** | **0.103** |
| | FNO | 0.1839 | 1.2 | 1.0342 | 1.4302 |
| | ORCA | 0.1818 | 1.0812 | 1.6316 | 1.6399 |
| 10 | UPS-B | **0.0107** | **0.3327** | **0.0134** | **0.0809** |
| | FNO | 0.1698 | 0.8193 | 0.67 | 0.567 |
| | ORCA | 0.1004 | 0.5376 | 0.4829 | 0.1623 |
| 100 | UPS-B | **0.0034** | 0.2508 | **0.0022** | **0.0543** |
| | FNO | 0.0037 | 0.1869 | 0.0123 | 0.3962 |
| | ORCA | 0.0051 | **0.1362** | 0.027 | 0.0898 |
| 9K (Full) | UPS-B | **0.0003** | **0.041** | **0.0008** | **0.0191** |
| | FNO | 0.0014 | 0.12 | 0.0031 | 0.098 |
| | ORCA | 0.0034 | 0.082 | 0.012 | 0.0287 |

shown in Table 2. As the number of adaptation samples increases, the prediction error decreases. Notably, the 100-shot result of UPS on 1D datasets is better than the baselines trained on 9,000 data, i.e., we use 90x less data to match the performance of single-family operators. This makes UPS useful for low-resource PDE problems where data collection is costly and training models from scratch is challenging. On 2D Diffusion-Reaction, we are slightly worse than pretrained MPP (0.0292) and DPOT (0.0106) since this dataset is considered as in-distrubution for MPP and DPOT.

**Unseen Coefficients** UPS also generalizes to PDEs in the same families as the training data but with different coefficients. We verify this by adapting UPS to Burgers Equation with $\nu = 1.0$ (the model is trained on $\nu = 0.001$) and compressible Navier-Stokes with $M = 1$, $\eta = \zeta = 0.1$ (the model is trained on $M = \eta = \zeta = 0.1$). The last two columns in Table 2 shows that while our zero-shot performance is already competitive, the performance after further adaptation outperforms most considered baselines.

**Unseen Resolutions** Zero-shot resolution refers to training the model on a lower resolution of the input data and evaluating them directly on a higher resolution. PDE solvers with this ability are better equipped to handle real-world scenarios where input data may vary in resolution due to practical constraints or sensor-based limitations. Recall that UPS is trained with $n$-point discretization $W_n^s$, and we set $n = 128$ because most 2D datasets in PDEBench has resolution 128. Now, we

Table 3: UPS with resolution 128 has an nRMSE of 0.0033 for Advection and 0.0931 for incompressible Navier-Stokes. We directly test UPS on higher resolutions.

| Test Resolution | 256 | 512 | 1024 |
|---|---|---|---|
| Advection (nRMSE) | 0.0057 | 0.0064 | 0.0068 |
| Incomp Navier-Stokes (nRMSE) | 0.119 | 0.126 | - |

evaluate the performance of UPS for $n \in \{256, 512, 1024\}$, increasing the resolution of the input PDE. This is achieved by downsampling the higher-resolution inputs to make them compatible with UPS and then upsampling the output prediction to the desired resolution. We do not fine-tune the model at all.

We report the resolution generalization performance for 1D Advection Equation and 2D incompressible Navier-Stokes in Table 3. Although the nRMSEs for both PDEs slightly increase compared to the nRMSE for the training resolution, they outperform all baselines in Table 1. Since the numbers are similar across columns, UPS generalizes to higher resolutions in a zero-shot manner.

## 5.3 Ablation Studies

We perform five sets of studies to ablate various design decisions in UPS. S1-S4 demonstrate why adapting from pretrained LLMs is beneficial, while S5 is related to the FNO embedding network.

**S1: Pretrained LLMs vs. Training From Scratch** Compared to existing single-family models like FNO, UPS uses a transformer-based architecture with more parameters and reuses the pretrained LLM weights for the model body. To show that our results are not solely attributed to the model size and that cross-modal adaptation is important, we evaluate the model's performance when we train a transformer model *from scratch* using the same PDE datasets without doing anything more complicated. As shown in Table 4, training from scratch results in much worse performance than UPS, showing the benefits of adapting a pretrained LLM.

**S2: Cross-Modal Alignment** We also test the importance of the two objectives used in stage 1, i.e., alignment loss with MMD, and task loss with nRMSE. We study three settings: (i) using only $\mathcal{L}_{\text{align}}$ for stage 1 as in Shen et al. (2023); (ii) using only $\mathcal{L}_{\text{task}}$ for stage 1; and (iii) removing stage 1 from our workflow entirely. As shown in Table 4, while removing any objective reduces the performance across all datasets, removing the task loss has a more significant negative effect. Meanwhile, removing the entire stage of embedding pretraining hurts prediction accuracy. This shows that simply fine-tuning the LLM without considering the modality gap or learning to extract PDE features is ineffective.

**S3: Incorporating Text-Form Metadata** UPS leverages the PDE's metadata by combining its text embeddings with the learned PDE embeddings. To study whether incorporating such metadata is helpful and identify an optimal approach, we compare our workflow with two alternatives: (i) we do not use metadata, so $h_{\text{mix}} := h_{\text{PDE}}$; (ii) we use metadata, but instead of concatenating features from two modalities, we apply a cross-attention mechanism: $h_{\text{mix}} := \text{softmax}(\frac{QK^T}{\sqrt{e}})V$, where $Q = W_Q h_{\text{PDE}}$, $K = W_K h_{\text{meta}}$,

Table 4: Results for the ablation studies. For each set of experiments, only the specified settings are different; all the other hyperparameters and training configurations are the same. Overall, the full UPS-Base workflow (first row for every study) most effectively leverages the pretrained knowledge of LLMs and obtains the best results.

| Study No. | Settings | Advection 1D | Burgers 1D | Diff-Sorp 1D | Navier-Stokes 1D | Shallow-Water 2D | Navier-Stokes 2D | Incomp Navier-Stokes 2D |
|---|---|---|---|---|---|---|---|---|
| S1 | Pretrained LLM | **0.0027** | **0.0399** | **0.0009** | **0.0056** | **0.0016** | **0.0153** | **0.0931** |
| | Training From Scratch | 0.017 | 0.0546 | 0.0036 | 0.0159 | 0.0032 | 0.0461 | 0.1442 |
| S2 | Align and Task | **0.0027** | 0.0399 | **0.0009** | **0.0056** | **0.0016** | **0.0153** | **0.0931** |
| | Task Only | 0.0048 | **0.0389** | **0.0009** | 0.0065 | 0.002 | 0.0184 | 0.1046 |
| | Align Only | 0.0039 | 0.043 | 0.0011 | 0.0063 | 0.0022 | 0.0187 | 0.1092 |
| | No Embedding Pretraining | 0.0049 | 0.0436 | 0.0019 | 0.0072 | 0.0024 | 0.0197 | 0.1079 |
| S3 | Concatenate Pretrained Text | **0.0027** | **0.0399** | **0.0009** | **0.0056** | **0.0016** | **0.0153** | **0.0931** |
| | Cross-Attention Pretrained Text | 0.003 | 0.0420 | **0.0009** | 0.0065 | 0.0023 | 0.0189 | 0.1082 |
| | Concatenate One-Hot | 0.0029 | 0.0447 | 0.0011 | 0.006 | 0.0018 | 0.0198 | 0.095 |
| | Concatenate Learned Embeddings | 0.0041 | 0.0474 | 0.0014 | 0.0119 | 0.0036 | 0.0295 | 0.1103 |
| | No Meta Information | 0.0122 | 0.0453 | 0.001 | 0.0091 | 0.0026 | 0.0238 | 0.1171 |
| S4 | RoBERTa-Base | **0.0027** | 0.0399 | **0.0009** | **0.0056** | **0.0016** | 0.0153 | 0.0931 |
| | Flan-T5-Base | 0.0094 | 0.0404 | 0.0076 | 0.0098 | 0.0028 | 0.037 | 0.1166 |
| | CLIP-Base | 0.0046 | **0.0321** | 0.0018 | 0.0063 | 0.0019 | **0.0151** | **0.0905** |
| S5 | $l = 32$ | 0.0027 | **0.0399** | **0.0009** | **0.0056** | **0.0016** | **0.0153** | **0.0931** |
| | $l = 20$ | **0.0024** | 0.0423 | **0.0009** | 0.0068 | 0.0022 | 0.0157 | 0.1043 |
| | $l = 8$ | 0.0032 | 0.0429 | **0.0009** | 0.0071 | 0.0024 | 0.0195 | 0.1064 |

and $V = W_V h_{\mathrm{meta}}$. To further investigate whether the pretrained text embeddings contribute beyond merely labeling the PDE type, we also study alternative embedding strategies that do not leverage language pretraining: (iii) we replace the pretrained text embeddings of the PDE meta information with one-hot encoded vectors representing each PDE type. This setting serve as a baseline to assess the impact of merely labeling the PDE types without any semantic understanding; (iv) we also test a setting where PDE types were embedded using a randomly initialized embedding layer that was trained from scratch along with the rest of the network, i.e., each new token represents a PDE family.

The results are shown in Table 4. UPS outperforms the non-metadata baseline, demonstrating the effect of incorporating metadata as a textual form of domain knowledge, which LLMs are able to understand. The results also suggest that feature concatenation is better than cross-modal attention, possibly because the latter is harder to optimize. Lastly, the setting utilizing pretrained text embeddings consistently outperforms the one-hot and embedding-from-scratch settings. In terms of learning dynamics, we also observe that using pretrained embeddings demonstrated faster convergence compared to the alternative strategies. This suggests that the pretrained semantic knowledge in the LLM indeed contributes to processing the PDE data, not just in labeling PDE types but might also in understanding the underlying physical phenomena. However, we leave studying the the exact mechanism of cross-modal transfer and the optimal combination of metadata and PDE data as a future direction.

**S4: Other LLMs/VLMs**  To study whether UPS applies to other pretrained models, we further investigate Flan-T5 (Chung et al., 2022) and the vision language model CLIP (Radford et al., 2021). In particular, for CLIP, we use its text model to encode the metadata and its vision model to process the PDE data. The results are reported in Table 4. Since these models are trained using the same datasets and optimizer configuration as RoBERTa, the results are not fully optimized. Nonetheless, their performance is competitive with existing baselines, and CLIP further outperforms RoBERTa on 3 tasks. This shows the compatibility of UPS with diverse pretrained backbones. A future direction is to study whether optimizing the training hyperparameters for each pretrained model—especially VLMs like CLIP that are trained for an additional vision modality—could improve downstream performance.

**S5: FNO Embedder & Target Sequence Length**  As discussed in Section 3, the channel $l$ of the FNO layers in the embedding network determines the sequence length of the PDE features that will be fed into the transformer layers. To study how this hyperparameter affects learning outcomes, we vary $l \in \{8, 20, 32\}$ and report the results in Table 4. In general, increasing $l$ improves the size and capacity of the embedding network, as well as the expressivity of the PDE features. This leads to lower prediction error. However, using too many parameters for the embedding network may result in a trade-off between effectiveness and efficiency. For instance, we also experimented with $l = 64$ (Appendix A.3.3) and find that the longer sequence length

leads to slight performance improvements but with much higher computational costs. Thus, we opt for $l = 32$ in our main experiments.

## 6 Conclusion and Future Work

In this paper, we present UPS, a method for adapting pretrained LLMs to unified time-evolution operators that predict the next state of a PDE from the current state. UPS applies to a diverse set of PDE families defined over one- and two-dimensional domains, with varying initial conditions, boundary conditions, coefficients, and resolutions. To train UPS, we develop a two-stage cross-modal adaptation protocol that first pretrains a FNO-based embedding network and aligns its hidden representations with the LLM's embedding space, and then fine-tunes the entire model on a dataset containing diverse families of PDEs. Since UPS is adapted from pretrained models, it requires fewer training samples and compute than previous approaches for training unified PDE solvers from scratch. We show that UPS achieves state-of-the-art performance across multiple datasets from PDEBench and is capable of zero- and few-shot transfer to different PDE families, coefficients, and resolutions.

We identify several future directions based on our work. First, we can validate our method on a broader range of PDEs with higher-order temporal derivatives or three-dimensional domains. Meanwhile, to seek a truly general foundation model for PDE, we aim to extend the types of tasks that UPS can solve. Currently, UPS is only applicable to forward prediction. It is important to study inverse problems of parameter estimation for different PDEs as well. For an impact statement, see Appendix A.5.

## Acknowledgement

We thank Mikhail Khodak and Wenduo Cheng for providing useful feedback on the paper. This work was supported in part by the National Science Foundation grants IIS1705121, IIS1838017, IIS2046613, IIS2112471, and funding from Meta, Morgan Stanley, Amazon, and Google. Any opinions, findings and conclusions or recommendations expressed in this material are those of the author(s) and do not necessarily reflect the views of any of these funding agencies.

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

# A  Appendix

## A.1  Datasets

As mentioned in Section 5, we train our models using the datasets provided in the PDEBench (Takamoto et al., 2022). The time-dependent PDE families considered by our models are: Burgers Equation (1D), Diffusion-Sportion (1D), Shallow-Water (2D), compressible Navier-Stokes (1D and 2D), incompressible Navier-Stokes (2D), and Diffusion-Reaction (1D and 2D). For each $s \in S$, the number of points in the $n$-point discretization $W_n^s$ is 128, i.e, $n = 128$. For PDEs where the PDEbench-provided grid has more than 128 points in each dimension, we sample 128 equispaced points.

In this section, we provide few key properties and considerations for the PDEs used in this paper. The initial conditions $\boldsymbol{u}(0, x)$ for most of the datasets are sampled from a superposition of sinusoidal waves. The set of coefficients and number of trajectories used per PDE are reported in Appendix Table 5. For full details on the data generation process and the hyperparameters used to generate the PDE dataset, we refer the reader to Takamoto et al. (2022).

### A.1.1  Burgers Equation (1D)

Burgers equation is commonly used to model the nonlinear dynamics of various fluid dynamics systems. Given the field $u(t, x) \in (0, 2] \times (0, 1) \to \mathbb{R}$ the PDE is defined as follows:

$$\partial_t u(t, x) + \partial_x \frac{u^2(t, x)}{2} = \frac{\nu}{\pi} \partial_{xx} u(t, x) \tag{5}$$

Here $\nu$ is the diffusion coefficient or the viscosity of the liquid, and $\pi$ is the density of the liquid.

### A.1.2  Diffusion-Sorption Equation (1D)

Diffusion-Sorption is a nonlinear diffusive process slowed down by an external force that is dependent of the state variable $u$ $R$. This PDE is used to model groundwater contamination transport processes. The PDE is defined as the following:

$$\partial_t u(t, x) = \frac{D}{R(u)} \partial_{xx} u(t, x), \tag{6}$$

where $x \in (0, 1)$, $t \in (0, 500]$, and $D = 5 \times 10^{-4}$. For more details on the initial conditions, boundary conditions and the function $R(u)$, we refer the reader to Takamoto et al. (2022). For our training, we use 4500 trajectories for this PDE generated by varying the initial conditions.

### A.1.3  Advection Equation (1D)

Given advection speed $\beta$, the advection equations are expressed as:

$$\begin{aligned} \partial_t u(t, x) + \beta \partial_x u(t, x) &= 0 \\ u(0, x) &= u_0(x) \end{aligned} \tag{7}$$

where $x \in (0, 1)$ and $t \in (0, 2]$. Various examples in this dataset are generated by sampling multiple initial conditions from a super-position of sinusoidal waves as used in Takamoto et al. (2022).

### A.1.4  Compressible Navier-Stokes (1D and 2D)

Given density $\rho$, velocity $\boldsymbol{u}$, pressure $p$, internal energy of the system $\epsilon$ the compressible Navier-Stokes equations are defined as follows.

$$\begin{aligned} &\partial_t p + \nabla \cdot (\rho \boldsymbol{u}) = 0, \\ &\rho \left( \partial_t \boldsymbol{u} + \boldsymbol{u} \cdot \nabla \boldsymbol{u} \right) = -\nabla p + \eta \Delta \boldsymbol{u} + \left( \xi + \frac{\eta}{3} \right) \nabla \left( \nabla \cdot \boldsymbol{u} \right) \\ &\partial_t \left( \epsilon + \rho \frac{\|\boldsymbol{u}\|_2^2}{2} \right) + \nabla \cdot \left( \left( p + \epsilon + \rho \frac{\boldsymbol{u}^2}{2} \right) \boldsymbol{u} - \boldsymbol{u} \cdot \sigma' \right) = 0 \end{aligned} \tag{8}$$

Here, $x \in (-1, 1)$ for 1D Navier-Stokes and $x \in (0, 1)^2$ for 2D Navier-Stokes, and $t \in (0, 1)$. Compressible Navier-Stokes stokes are used to model multiple real-world phenomena in aerodynamics and fluid dynamics.

### A.1.5    Incompressible Fluid Navier-Stokes (2D)

We define the equations for incompressible fluid Navier-Stokes where we impose the condition that the fluid is "incommpressible." That is, the equation follows the following condition:

$$\nabla \cdot \boldsymbol{u} = 0 \tag{9}$$

For density $\rho$ and pressure $p$, the equations used to generate the data in Takamoto et al. (2022) are as follows:

$$\rho \left( \partial_t \boldsymbol{u} + \boldsymbol{u} \cdot \nabla \boldsymbol{u} \right) = -\nabla p \boldsymbol{u} + \eta \Delta \boldsymbol{u} + \boldsymbol{f} \tag{10}$$

where $\boldsymbol{f}$ is an external forcing function, and Dirichlet boundary conditions. Here $x \in [0, 1]^2$ and the initial conditions $\boldsymbol{u}$ and the forcing term $\boldsymbol{f}$ are sampled from two-dimensional Gaussian random fields. Please refer to Takamoto et al. (2022) for more details on the data generation process.

### A.1.6    Reaction Diffusion (1D and 2D)

Reaction Diffusion are diffusive processes with external force applied to the system that may or may not depend over the field variable $\boldsymbol{u}$. They are often used to model many thermodynamical systems.

1D reaction diffusion is defined as follows:

$$\partial_t u(t, x) - \nu \partial_{xx} u(t, x) = \rho u(t, x)(1 - u(t, x)) \tag{11}$$

for all $x \in (0, 1)$ and $t \in (0, 1]$.

For 2D reaction diffusion, let $\boldsymbol{u}(t, x) = [u_1(t, x), u_2(t, x)]$. Then the equations are defined as:

$$\begin{aligned}
\partial_t u_1(t, x) &= \nu_1 \partial_{x_1 x_1} u_1 + \nu_1 \partial_{x_2 x_2} u_1 + u_1 - u_1^3 - k - u_2 \\
\partial_t u_1(t, x) &= \nu_2 \partial_{x_1 x_1} u_2 + \nu_2 \partial_{x_2 x_2} u_2 + u_1 - u_2
\end{aligned} \tag{12}$$

where $k = 5 \times 10^{-3}$ and $\nu_1$ and $\nu_2$ are diffusion coefficients. Here $x_1 \in (-1, 1)$ and $x_2 \in (-1, 1)$ and the initial conditions are sampled from a Gaussian random field.

### A.1.7    Shallow-Water Equations (2D)

These are derived from Navier-Stokes and are a framework for modelling free-surface flow problems. We denote by $u_1(x)$, and $u_2(x)$ as the velocities in the horizontal and vertical directions and $h$ as the height of the water and $b$ defining the spatially varying bathymetry (the measurement of the depth of water in oceans, rivers, or lakes). The shallow-water equations are defined as follows:

$$\begin{aligned}
&\partial_t h + \partial_{x_1} h u_1 + \partial_{x_2} h u_2 = 0, \\
&\partial_t h u_1 + \partial_{x_1} \left( u_1^2 h + \frac{1}{2} g_r h^2 \right) + \partial_{x_2} u_1 u_2 h = -g_r h \partial_{x_1} b, \\
&\partial_t h u_2 + \partial_{x_2} \left( u_2^2 h + \frac{1}{2} g_r h^2 \right) + \partial_{x_1} u_1 u_2 h = -g_r h \partial_{x_2} b,
\end{aligned} \tag{13}$$

where $x \in [-2.5, 2.5]^2$ and $g_r$ is the gravitational acceleration.

### A.1.8    Summary

The following table summarizes the coefficients of the datasets used to train and test our model (note that 1D/2D Diffusion-Reaction only appear in the test set but not the training set). We also provide the number of training and test trajectories. We generate the input-output pairs using autoregressive teacher-forcing.

Table 5: For each PDE family, we select one set of coefficients and use the data for training and testing UPS.

| Dimension | Dataset | Coefficients | Num Train Trajectories | Num Test Trajectories | Timesteps | Resolution |
|---|---|---|---|---|---|---|
| | Advection | $\beta = 0.4$ | 4500 | 1000 | 41 | 128 |
| | Burgers | $\nu = 0.001$ | 4500 | 1000 | 41 | 128 |
| 1D | Diffusion-Reaction | $\nu = 0.5, \rho = 1.0$ | 4500 | 1000 | 21 | 128 |
| | Diffusion-Sorption | - | 4050 | 100 | 21 | 128 |
| | Compressible Navier-Stokes | $\eta = \zeta = 0.1$, rand_periodic | 4500 | 1000 | 21 | 128 |
| | Shallow-Water | - | 405 | 10 | 101 | 128 |
| 2D | Diffusion-Reaction | - | 405 | 10 | 101 | 128 |
| | Compressible Navier-Stokes | $M = \eta = \zeta = 0.1$, periodic | 4500 | 1000 | 21 | 128 |
| | Incompressible Navier-Stokes | $M = 0.1, \eta = \zeta = 1E - 8$ | 4500 | 1000 | 21 | 128 |

## A.2 Experiment Details

### A.2.1 Training Hyperparameters

We use the following training hyperparameters for all of our experiments, unless otherwise specified. Due to time constraint, we have not performed exhausitive hyperparameter search or tailor the hyperparameters to each experiment setting.

- Batch size: 32

- Gradient accumulation: 1

- Gradient clipping: -1

- Dropout: 0

- Optimizer: Adam

- Learning rate: 5E-5

- Weight decay: 1E-5

- Training epoch: 20 for stage 1, 100 for stage 2

We use the CoNLL-2003 dataset (Sang & Meulder, 2003) as the reference dataset for alignment in stage 1.

### A.2.2 Efficiency Analysis

We run all of our experiments on a single NVIDIA A6000. Table 6 show the detailed model size, per epoch training time (in seconds), and total training time (in hours) for different network configurations. Note that we train the models for 100 epochs.

Table 6: Trainable parameters and training time for each LLM backbone.

| | RoBERTa-Base | RoBERTa-Large | Flan-T5-Base | CLIP-Base |
|---|---|---|---|---|
| Num Params | 149M | 387M | 176M | 132M |
| Per Epoch (s) | 3200 | 7600 | 3500 | 3000 |
| Total (hrs) | 88 | 211 | 97 | 83 |

We reported additional metrics such as FLOPs and the time required for predicting a single step for a PDE instance in Table 7, assuming the input data is 2D with 4 channels and resolution 128. We mainly compared with unified models that have similar model sizes. Compared to these existing work, UPS has lower FLOPs and shorter inference time. This shows that our model is ideal for deployment in practical environments where both computational efficiency and speed are critical.

Table 7: Efficiency comparison for unified neural operators.

|  | UPS-B | MPP-B | DPOT-M |
|---|---|---|---|
| Num Params | 149M | 116M | 122M |
| Per Forward Pass FLOPs (G) | 72.66 | 102.12 | 75.44 |
| Single Step Inference Time (ms) | 1.77 | 2.34 | 1.88 |

### A.3  Detailed Experiment Results

### A.3.1  2D-Only UPS

Table 8: Training UPS with all of the 2D datasets in PDEBench and compare with MPP and DPOT. Note that beyond these PDEBench datasets, MPP is also pretrained on PDEArena (Gupta & Brandstetter, 2022) and DPOT is pretrained on PDEArena (Gupta & Brandstetter, 2022) as well as CFDBench (Yining et al., 2023). Baseline results taken from Hao et al. (2024). '-' means that the result is not available.

|  |  | # Params (sorted within groups) | PDEBench 2D Navier Stokes-$(\eta, \zeta)$ | | | | | | 2D Diff-React | 2D Shallow-Water |
|---|---|---|---|---|---|---|---|---|---|---|
|  |  |  | 1,0.1 | 1,0.01 | M1 | 0.1,0.1 | 0.1,0.01 | M0.1 |  |  |
| Small-Sized | FNO | 0.5M | 0.098 | 0.096 | 0.097 | 0.360 | 0.170 | 0.265 | 0.12 | 0.0044 |
|  | FFNO | 1.3M | 0.0212 | 0.052 | 0.0366 | 0.162 | 0.0452 | 0.104 | 0.0571 | 0.0116 |
|  | GNOT | 1.8M | 0.0325 | 0.0420 | 0.0373 | 0.0228 | 0.0341 | 0.0285 | 0.0311 | 0.00678 |
|  | Oformer | 1.9M | 0.0417 | 0.0625 | 0.0521 | 0.0254 | 0.0205 | 0.0229 | 0.0192 | 0.00717 |
| Medium-Sized | MPP-Ti | 7M | – | – | 0.0442 | – | – | 0.0312 | 0.0168 | 0.0066 |
|  | DPOT-Ti | 7M | 0.0173 | 0.0397 | 0.0285 | 0.0132 | 0.0220 | 0.0176 | 0.0321 | 0.00560 |
|  | MPP-S | 30M | – | – | 0.0319 | – | – | 0.0213 | **0.0112** | 0.0024 |
|  | DPOT-S | 30M | 0.0153 | 0.0337 | 0.0245 | 0.0119 | 0.0187 | 0.0153 | 0.0379 | 0.00657 |
|  | DPOT-M | 122M | 0.0116 | **0.0238** | **0.0177** | 0.00866 | 0.0129 | **0.0108** | 0.0292 | 0.0029 |
|  | **UPS-B (Ours)** | 149M | **0.0112** | 0.0605 | 0.0277 | **0.0085** | **0.0124** | 0.0211 | 0.0243 | **0.0018** |
| Large-Sized | **UPS-L (Ours)** | 387M | 0.0102 | 0.0596 | 0.024 | **0.0083** | **0.0102** | 0.0209 | 0.0236 | **0.0015** |
|  | MPP-L | 400M | – | – | 0.0208 | – | – | 0.0147 | **0.0098** | 0.00220 |
|  | DPOT-L | 500M | 0.0100 | 0.0216 | 0.0158 | 0.00872 | 0.0115 | 0.0101 | 0.0232 | 0.00233 |
|  | DPOT-H | 1.03B | **0.00961** | **0.0180** | **0.0138** | 0.00847 | 0.0105 | **0.00948** | 0.0191 | 0.00199 |

### A.3.2  Few-Shot Adaptation

Compared to full fine-tuning of stage 2, we lower the learning rate when performing few-shot adaptation to prevent catastrophic forgetting.

- Batch size: 32

- Gradient accumulation: 1

- Gradient clipping: -1

- Dropout: 0

- Optimizer: Adam

- Learning rate: 1E-5

- Weight decay: 1E-5

- Epoch: 100

The following table reports the time required for few-shot experiments. Note that for Burgers equation, we train the model using $\nu = 0.001$, but the results here are for $\nu = 1.0$.

Table 9: Time for few-shot experiments. Our model outperforms most existing baselines on these tasks by using fewer than 500 samples and much shorter adaptation time.

| Num Samples | 1D Diffusion-Reaction | | 2D Diffusion-Reaction | | Burgers $\nu = 1.0$ | |
|---|---|---|---|---|---|---|
| | Per Epoch (s) | Total (hrs) | Per Epoch (s) | Total (hrs) | Per Epoch (s) | Total (hrs) |
| 10 | 2 | 0.05 | 12 | 0.33 | 3 | 0.08 |
| 50 | 10 | 0.28 | 48 | 1.33 | 10 | 0.28 |
| 100 | 23 | 0.64 | 112 | 3.11 | 40 | 1.11 |
| 500 | 112 | 3.11 | 512 | 14.22 | 96 | 2.67 |

### A.3.3 Ablation on Longer Sequence Length

We studied the effect of embedding sequence length in Section 5.3 paragraph S5 of the main paper. The results show that among $l = \{8, 20, 32\}$, larger $l$ indeed leads to better performance. However, since LLMs can support sequence lengths much longer than $l = 32$, we consider expanding the feature length (the number of "tokens") used to represent PDE data. See results below.

| | Advection 1D | Burgers 1D | Diffusion-Sorption 1D | Navier-Stokes 1D | Shallow-Water 2D | Navier-Stokes 2D | Incomp Navier-Stokes 2D |
|---|---|---|---|---|---|---|---|
| $l = 32$ | **0.0027** | 0.0399 | **0.0009** | 0.0056 | 0.0016 | **0.0153** | **0.0931** |
| $l = 64$ | 0.0034 | **0.038** | **0.0009** | **0.0054** | **0.0015** | 0.0162 | 0.0988 |

While $l = 64$ performs slightly better on some tasks, increasing the sequence length means that (i) the embedding network is going to be larger (since $l$ also corresponds to the width of the FNO layers), and (ii) the training time will increase as each sequence is longer. Both increase the training cost. Hence, we want to select the $l$ that achieves a balance between efficiency and effectiveness. That's why we use $l = 32$ for our main experiments.

### A.3.4 Long-Horizon Prediction

As stated in Section 3, our method mainly focuses on predicting the next step from the current step, i.e., $\hat{\boldsymbol{u}}_{t+1}^s(\boldsymbol{x}) = \mathcal{G}_\theta(\boldsymbol{u}_t^s(\boldsymbol{x}))$. However, we are also interested in the prediction capacity of our method over a longer period of time. Thus, we study an additional setting that predicts $\hat{\boldsymbol{u}}_{t+10}^s(\boldsymbol{x})$. We show non-autogressive evaluation results since otherwise we will only have very few time steps for each test PDE trajectory. The table below shows that UPS is still effective for long-horizon prediction compared to baselines like FNO. Even though the prediction interval is longer, the error rates only slightly increase possible because we use non-autoregressive evaluation, so the errors do not accumulate.

| | Advection 1D | Burgers 1D | Diffusion-Sorption 1D | Navier-Stokes 1D | Shallow-Water 2D | Navier-Stokes 2D | Incomp Navier-Stokes 2D |
|---|---|---|---|---|---|---|---|
| $\Delta t = 1$ | 0.0027 | 0.0399 | 0.0009 | 0.0056 | 0.0016 | 0.0153 | 0.0931 |
| $\Delta t = 10$ | 0.0034 | 0.04 | 0.0011 | 0.0074 | 0.0026 | 0.0189 | 0.134 |

## A.4 Visualization

### A.4.1 Burgers Equation

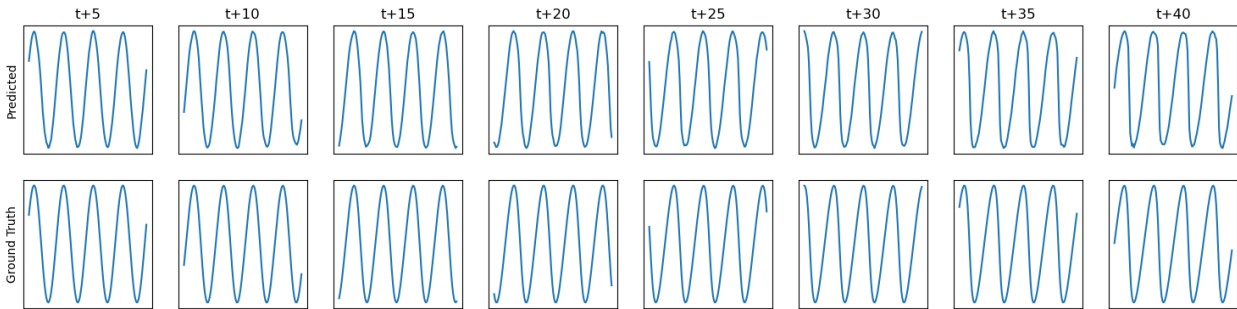

### A.4.2 Diffusion-Sorption

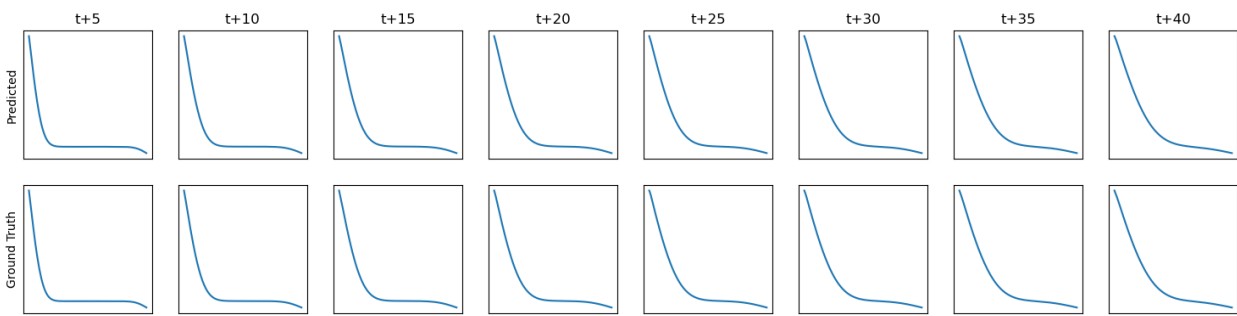

### A.4.3 1D Navier Stokes

We show $V_x$, density, and pressure.

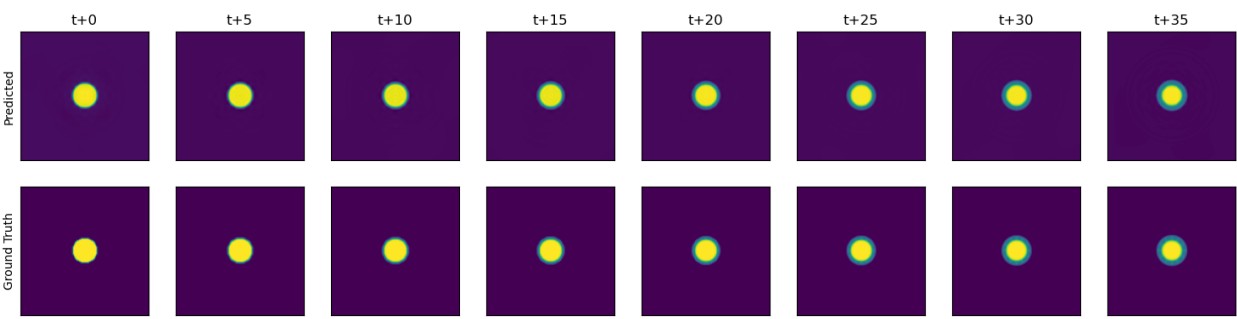

### A.4.4 Shallow Water

### A.4.5 2D Navier Stokes

We show $V_x$, $V_y$, density, and pressure.

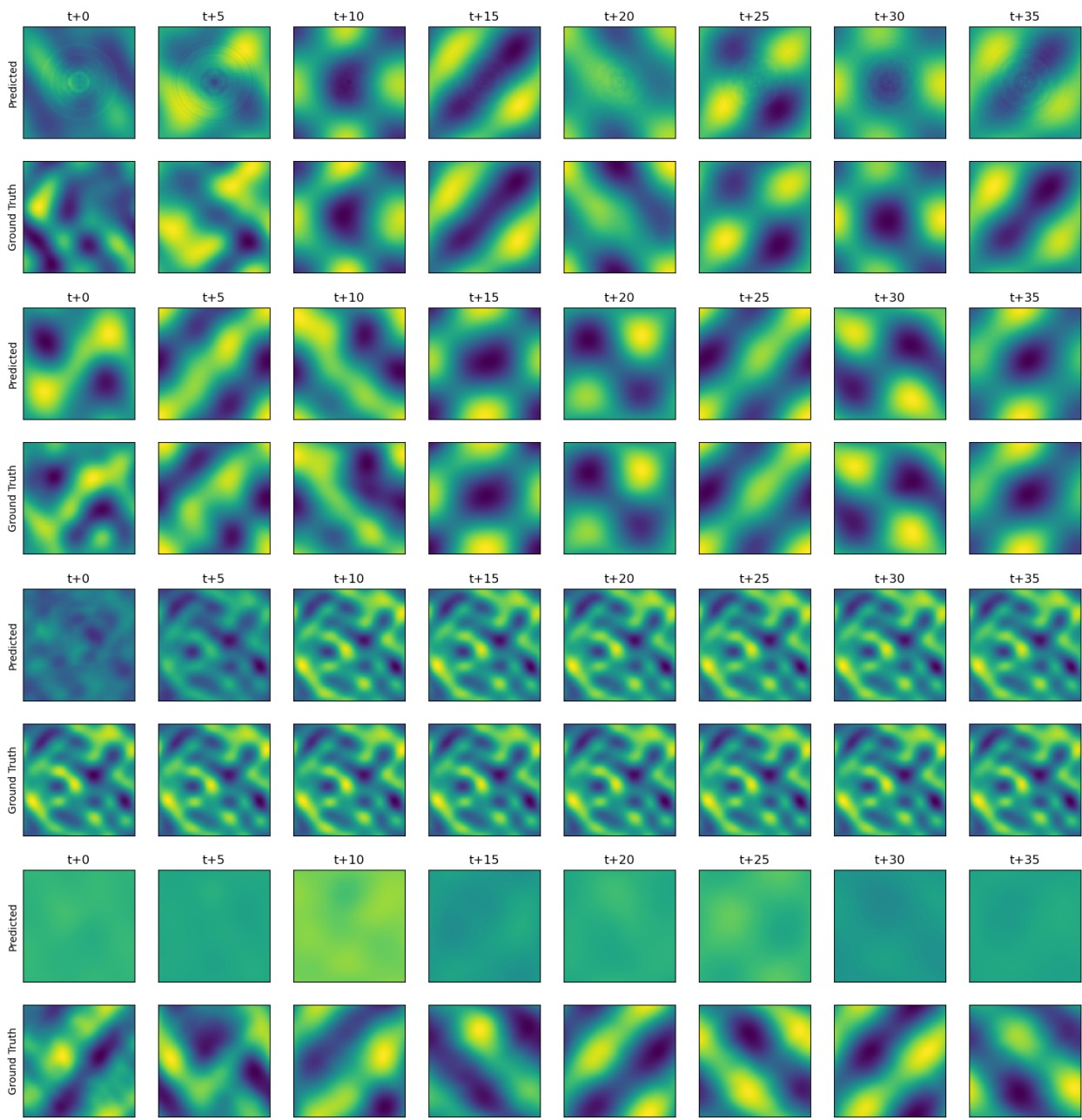

In the prediction for 2D compressible Navier-Stokes we see a few artifacts in our generation. Furthermore, for quantities like pressure, our network often seems to generate an overly smoothened output. This could be because the 2D Navier-Stokes is the only PDE in our dataset that requires us to model pressure, and therefore the network is biased towards predicting a uniform value, which in our case is 0. We believe this can be avoided by adding more families of PDEs that model pressure, and is a fertile ground for future work.

### A.5 Broader Impact

This paper calls for ML community's attention to take advantage of LLMs and apply them to a wider range of real-world problems beyond the NLP domains. This moves towards truly democratizing machine learning in real life. In terms of broader societal impact, our work can exert a positive influence as it contributes to reusing existing models and resources, reducing the computational burden of developing new large-scale models on massive data. However, lowering the barrier for applying LLMs to a wide range of tasks necessarily comes with the risk of misuse. Hence, it is imperative to develop adaptation methods with better privacy, safety, and fairness guarantees.

