# OpenReview forum: "UPS: Efficiently Building Foundation Models for PDE Solving via Cross-Modal Adaptation"
_TMLR — Accepted by TMLR_

### Review · Reviewer_WV3b · 2024-08-30

**Summary Of Contributions:**

The paper presents a method for solving PDEs using a transformer-based architecture, which is pretrained on a large language model (LLM). The method uses an FNO network for mapping the PDE inputs into a latent space which is aligned with the LLM data distribution. In experiments, the method is trained across multiple PDE families and is presented as being more efficient in terms of data and compute. Evaluated on PDE families from PDEBench, it achieves competitive results with other existing methods.

**Audience:**

Yes

**Broader Impact Concerns:**

Sufficient impact statement provided in A.6.

**Claims And Evidence:**

Yes

**Requested Changes:**

*Critical*:
- Notation in Section 3.1 needs to be carefully checked and made consistent. The spatial variable is defined as $\mathbf{x}\in\Omega^s\subset \mathbb{R}^{d_s}$, while shortly after, $x_i^s\in\mathbb{R}^{d_s}$ is defined. Why is $\mathbf{x}$ in a subset of $\mathbb{R}^{d_s}$, while $x_i^s$ is not? And why is $\mathbf{x}$ bolded while $x_i^s$ is not, although both have the same dimensionality? This should be clarified. The notation of $u^s_t(\mathbf{x})$ for all values of $x_i^s$ is also confusing and should be better separated.
- Section 3.1: "We assume that n is constant across PDE families." - This statement would need an explanation on why this is a reasonable assumption. In particular with varying spatial dimensions, this might not be the case, and some more chaotic PDEs require finer discretizations than others.
- Section 3.2, FNO Embedding Network and Utilizing Pretrained LLMs: it seems unintuitive to map the channels of the FNO to the sequence length, instead of using a spatial discretization, as for instance in Vision Transformers. Similarly, in the output prediction, a linear layer maps the latent space of the LLM directly to the full spatial output dimensionality. As this design choice is very unintuitive, it needs be motivated in more detail. Further, in terms of number of tokens, what is expected to be represented by each token? How do they differ, and how are attention layers helping in this context?

*Would strengthen the work*:
- The generalization tests to unseen PDEs is limited. If the model is so training efficient, it would make the paper more convincing by doing a cross-validation style of generalization test, where the model is trained on a subset of PDEs and tested on the rest.

*Minor changes*:
- Page 1, top: "20,000 training data," -> "20,000 training data points,"
- Page 4, top: "These PDEs could model a range quantities" -> "These PDEs could model a range of quantities"
- Page 4, Section 3.1: "each u[...] is solution to a PDE" -> "each u[...] is a solution to a PDE"
- Page 4, Section 3.1: "the value of the the function" -> "the value of the function"

**Strengths And Weaknesses:**

### Strengths

The paper presents a novel approach to solving PDEs using a transformer-based architecture, which is pretrained on a large language model. This is an interesting approach in re-using pretrained language models in the context of PDE solving.

The method is shown to be more efficient in terms of data and compute compared to existing methods. This is an important aspect, in particular under the light of the reused LLM backbones. The method is applied across a variety of PDE families and achieves strong results on PDEBench.

The experiments on unseen PDE families is an interesting aspect of the paper, showing that the method can be transferred to new PDE families with few-shot learning. This is an important feature for practical applications, especially for a generalization to new PDEs with potentially limited data or new conditions.

The paper is clearly structured and uses established benchmarks. The experiments are well-documented and the results are presented in a clear and understandable way.

### Weaknesses

While the method has convincing empirical results, the theoretical foundation of the method is unintuitive and not well studied. As LLMs have been trained on very different data, it is unclear why they should be useful for PDE solving. In particular, this is in contrast to structured Transformer architectures for 2D data, like Vision Transformers, which could be directly applied to the observed data. The paper should provide a more detailed motivation for the usage of LLMs in this context, as well as how these tokens are used in the LLM. For more detailed comments, see requested changes.

Further, while the method uses language models, the focus is on encoder-based models which is not the most recent development in the field. The paper should discuss the choice of the transformer architecture in more detail, and how the method could work on autoregressive models, which are more commonly used in the context of LLMs nowadays.

Additionally, the paper highlights the choice of the FNO network for being resolution-invariant. However, the whole backbone with the embedding network is not resolution-invariant and fixes the spatial dimensions in its linear layers. This needs to be clarified in the paper.

---

> ### Author Response · Authors · 2024-10-15
> **Updated Manuscript and Author Response**
>
> Thank you for your review! Please find our response below.
>
> &nbsp;
>
> **(Critical) "Notation in Section 3.1 needs to be carefully checked and made consistent."**
>
> Thank you for pointing out this inconsistency in our notation. We have corrected the notation in Section 3.1 and have now bolded all $x_i^s$ in the paper appropriately.
>
> &nbsp;
>
> **(Critical) "'We assume that $n$ is constant across PDE families.' This statement would need an explanation on why this is a reasonable assumption. In particular with varying spatial dimensions, this might not be the case, and some more chaotic PDEs require finer discretizations than others."**
>
> The choice of $n$ reflects a carefully considered design decision aimed at optimizing performance across the diverse range of PDEs studied in our work. This value represents the minimum that consistently yields robust results across all cases.
> It is worth noting that we operate within the neural operator framework, where ideally, the number of training points should not significantly impact model performance. One of the key advantages of this approach is its potential to achieve accurate results with fewer discretization points compared to traditional numerical solvers. While we have not yet fully realized this ideal scenario, we believe it represents a promising and important direction for future research. Expanding our datasets and utilizing larger models may further enhance our ability to achieve these goals.
>
> &nbsp;
>
> **(Critical) "FNO Embedding Network and Utilizing Pretrained LLMs."**
>
> Our initial design choice to map the FNO channels to the sequence length was driven by the goal of modeling cross-channel information within the FNO features. However, we acknowledge that employing spatial discretization could potentially enhance the model’s ability to more explicitly capture and utilize spatial relationships. We intend to explore this architectural modification in our future research.
>
> &nbsp;
>
> **(Would strengthen the work) "The generalization tests to unseen PDEs is limited. If the model is so training efficient, it would make the paper more convincing by doing a cross-validation style of generalization test, where the model is trained on a subset of PDEs and tested on the rest."**
>
> Thank you for your constructive feedback. To enhance the generalizability results in the paper, we further performed two sets of experiments. First, for few-shot generalization, we used the another challenging dataset, 2D compressible Navier-Stokes equations with $M=1$, $\eta = \zeta=0.1$*, to better showcase our model's performance:
>
> |Num Adaptation Samples | 0|10  |100|Full|
> |:-:|:-:|:-:|:-:|:-:|
> |FNO|1.4302|0.567|0.3962|0.098|
> |ORCA|1.6399|0.1623|0.0898|0.0287|
> |UPS|0.0566|0.0134	|0.0039|0.0022|0.0009|
>
> These results show that UPS can generalize to challenging PDEs with limited training examples after being fine-tuned on PDE data from multiple families.
>
> Second, for zero-shot super-resolution, we added incompressible Navier-Stokes as well. Note that the PDEBench dataset has a maximum resolution of 512.
>
> | Test Resolution             | 256   | 512   |
> |-----------------------------|-------|-------|
> | Incomp Navier-Stokes (nRMSE)| 0.119 | 0.126 |
>
> These additional results and detailed analysis can be found in Table 2 and Table 3 of the revised manuscript. While we agree that a cross-validation style of generalization test could significantly enhance our model’s validation, this approach entails substantial computational costs due to the need to train a separate model for each test dataset. Consequently, we leave providing a comprehensive cross-validation assessment of the model's generalization capabilities as an important future step.
>
> &nbsp;
>
> **(Minor changes)**
>
> We have revised the manuscript to reflect the "minor changes". Specifically:
> - Changed  "20,000 training data" to "20,000 training trajectories" on page 3.
> - Changed  "These PDEs could model a range quantities" to "These PDEs could model a range of quantities" on page 4.
> - Changed "each u[...] is solution to a PDE" to "each u[...] is a solution to a PDE" on page 4.
> - Changed "the value of the the function" to "the value of the function" on page 4.

---

> ### Author Response · Authors · 2024-10-15
> **Updated Manuscript and Author Response (Cont.)**
>
> **Regarding the weakness, "while the method has convincing empirical results, the theoretical foundation of the method is unintuitive and not well studied. As LLMs have been trained on very different data, it is unclear why they should be useful for PDE solving."**
>
> We understand your concerns about the intuitiveness and theoretical rationale behind leveraging LLMs for PDE solving. Here we provide some ideas that might shed light on these inquiries.
>
> **Better initialization:** A key advantage of using LLMs is that we start from a meaningful initialization rather than a random one for fine-tuning.  This could potentially lead to quicker convergence and better ability to represent the complex solution spaces inherent in PDEs. That is, the model requires fewer iterations and less data to fine-tune for different types of PDEs. While this is our hypothesis, the experiments in Table 4 S1 provided empirical support, since the fine-tuned UPS performs much better than training the same architecture from scratch.
>
> **Leveraging pretrained embeddings:** Our framework also uses the text embedding of PDE families and coefficients for each data sample. These pretrained embeddings can effectively encode and categorize different types of PDEs, which  is helpful for handling multiple PDE families within a unified model framework. To further support this, we have performed addition experiments that encode the meta data without using the pretrained LLM (see Table 4 S3 of the revised paper). The fact that these settings underperform the text-embedding setting shows that leveraging the pretrained knowledge of LLMs can be beneficial.
>
> That being said, we acknowledge that the theoretical foundation of cross-modal transfer could be further strengthened. Our future work aims to explore more deeply the properties of LLMs (and more generally, pretrained models) that might contribute to PDE solving.

---

### Review · Reviewer_7gUR · 2024-09-09

**Summary Of Contributions:**

This paper proposes a new foundation model (FM) for solving partial differential equations (PDEs). Different from the previous FM for PDEs work, this paper claims to achieve better accuracy in a more efficient way on a benchmark dataset by (1) a new unified data representation scheme and (2) a unified network structure. In particular, the unified network structure includes combining several LLM blocks in order to take into account the PDE and its parameter information and feature improvement. The experiment indicates that the usage of the pre-trained LLM improves the performance even for the present PDE tasks.

**Audience:**

Yes

**Broader Impact Concerns:**

The broader impact statement is provided in Section A.6 and there seems little concerns on the ethcal implications because of the treating topic (pure scientific application for the moment).

**Claims And Evidence:**

No

**Requested Changes:**

Major:

* W1: Properly describe the evaluation method: 1-step prediction, not autoregressive, with the justification of why not evaluating autoregressively. If possible, evaluation in an autoregressive manner and 1-step prediction but far future time-step are desirable to improve the paper value.

* W2: Re-assess the computational cost using the same setting for all the models. Assessing the computational cost with flops would also help to improve the paper value.

* W3: Reconsider using a more difficult dataset for resolution and zero/few-shot study to evaluate the model performance more appropriately.

Minor:

* W4: Approach W4.1 – W4.4 appropriately.

* W5: Approach W5.1 -- W5.3 appropriately.

Besides,

* Eq. 13 (Shallow-Water Equations 2D):  Seemingly, the perpendicular direction’s derivative terms are missing: $\partial_{x2} (u_1 u_2 h)$ in the second equation and $\partial_{x1} (u_1 u_2 h)$ in the third equation.

* Please check if using color font (Figure 2 caption) is allowed, which is rarely seen in scientific journals.

**Strengths And Weaknesses:**

Strength:

* S1: The model performance is consistently better than the previous work.

* S2: The ablation study partially supports the effectiveness of each component of the model, in particular, the effectiveness of the pre-trained LLM as the encoder block.

* S3: Possibly, the model might be the most data-efficient method, though this claim needs more analysis as described below.


Weaknesses:

W1: Evaluation:
As far as I checked in the code (main.py L196, (function: evaluate)), the evaluation is a so-called “1-step” prediction, not an autoregressive prediction (the previous time-step prediction is used to predict the next time-step), even for the inference. This does not seem properly emphasized in the main body, such as in the experiment section. It could be highly possible that the cited baseline performance would be partly measured in autoregressively. So, the authors should clarify that the other baselines are also measured in a one-step prediction manner. Moreover, the evaluation of the autoregressive prediction is directly related to the real-world application (emulating numerical simulation and evaluating long-time behavior). So, the absence of an autoregressive evaluation reduces the value of this paper. One possibility to make the one-step prediction method more appropriate for a realistic application is to predict a far future step (10 to 100 steps of future information), not one time-step future which is usually a very easy task for ML models.

W2: Efficiency analysis:
Table 1 and efficiency analysis seem scientifically meaningless. As far as I understood, the total computation cost and the error are not evaluated using the same training and test dataset but citing the original paper’s numeric information. Although one can speculate something from it, it cannot be regarded as a scientific analysis. If the authors want to claim Table 1 as their major result, the comparison should be performed more carefully and correctly (using the same datasets, training setup, and hopefully similar model size).
Besides, it would be helpful for readers to measure the computational cost in terms of flops of the model.

W3: Using a too-easy dataset to evaluate the performance (resolution & zero/few-shot study):
For the resolution study, the 1D advection equation is too easy. It is helpful for readers to evaluate using more difficult datasets, such as 2D NS equations with figures of the predictions.
For zero/few-shot studies, diffusion reaction equations are too easy. It would be fruitful to evaluate them with alternative datasets, such as the Kuramoto-Sivashinsky 1D equation and 2D compressible NS (M=1 or larger).

W4: Insufficient explanations:
There are several descriptions where only insufficient or no information is provided (though I might miss them during the reviewing process). For example,

* W4.1: for $h_{LM}$, the explanation of the external reference NLP dataset is missing.

* W4.2: Is the linear predictor shared for all the PDEs or prepared for each PDE?

* W4.3: In section 5, the authors wrote that the dataset is normalized, which results in the solution not satisfying the original PDEs if considering a non-linear equation (e.g., consider $\partial u/\partial t + u \partial u/\partial x = 0$. Assuming $u_0$ is a solution of the equation, $k u_0$ ($k$ is a constant) does not satisfy the equation:  $k \partial u_0/\partial t + k^2 u_0 \partial u_0/\partial x = k (\partial u_0/\partial t + k u_0 \partial u_0/\partial x) \neq 0)$, where the $k$ can be introduced by the normalization. If the authors claim that their model is a neural operator, they should validate why they are allowed to perform the data normalization which actually alters the considering PDEs.

* W4.4: There are several undefined symbols, such as $d, d_u, d_s, d^s_u$ (Sec. 3.1), (x, 0) (Sec 3.1 Unifying Dimension, this usage of the open parenthesis appears only here), and UPS-B (Section 5.2).

W5: Some description needs to be polished:

* W5.1: In section 3, the label and model prediction are unified as $u^s_t$, which is very confusing to distinguish them. (moreover, sometimes $u^s_t$ becomes $u^t_s$, which increases the confusion).

* W5.2: Equation (1)’s L is not an operator but seems a function. Maybe
     $$\hat{L} (I, \nabla, \nabla \nabla, \cdots) u$$
    would be more appropriate as a description of an operator, mathematically.
   Besides, $B(u) = 0$, in Eq. (1) means only considering the Dirichlet boundary condition. Is it the authors’ intention (if we consistently apply the description of the original L)?

* W5.3: The English of the paper should be polished, e.g., verbs are sometimes used as nouns.

---

> ### Author Response · Authors · 2024-10-15
> **Updated Manuscript and Author Response**
>
> Thank you for your review! Please find our response below.
>
> &nbsp;
>
> **(Major W1) "Properly describe the evaluation method: 1-step prediction, not autoregressive, with the justification of why not evaluating autoregressively. If possible, evaluation in an autoregressive manner and 1-step prediction but far future time-step are desirable to improve the paper value...One possibility to make the one-step prediction method more appropriate for a realistic application is to predict a far future step (10 to 100 steps of future information), not one time-step future which is usually a very easy task for ML models."**
>
> Thank you for your comments regarding the clarity of our evaluation methodology. Following the PDEBench setup, we indeed used autoregressive evaluation for the results reported in the paper. The evaluation function in `main.py` is primarily used for monitoring validation loss during fine-tuning, as single-step evaluation is more computationally efficient than autoregressive evaluation due to batching. We have updated the supplementary material to include the autoregressive evaluation script. We have also revised the first paragraph of Section 5 to clarify the nature of our evaluation.
>
> Following your suggestion, we also added an experiment where the model uses $u_t$ to predict $u_{t+10}$. We retrained an RoBERTa-Base model, and the performance is shown below. These results are non-autogressive since otherwise we will only have very few time steps for each test PDE trajectory.
>
> | |Autoregressive? |1D Advection  |1D Burgers |1D Diffusion-Sorption   | 1D Comp Navier-Stokes  |  2D   Shallow-Water   | 2D Comp Navier-Stokes | 2D Incomp Navier-Stokes |
> |:-:|:-:|:-:|:-:|:-:|:-:|:-:|:-:|:-:|
> |FNO|Yes| 0.011 |0.042 |0.0017 |0.068 |0.0044 |0.36 |0.0942|
> |$\Delta t=1$ |Yes| 0.0027 |0.0399 |0.0009 |0.0056 |0.0019 |0.0153 |0.0931|
> | $\Delta t=10$|No |0.0034 |0.04 |0.0011 |0.0074 |0.0026 |0.0189 |0.134|
>
> These results showed that UPS is still effective for long-horizon prediction compared to baselines. For many PDEs like Burgers and Advection, even though the prediction interval is longer, the error rates only slightly increase, possibly because we use non-autoregressive evaluation, so the errors do not accumulate. We have added these results to Appendix A4.4 of the paper.
>
> &nbsp;
>
> **(Major W2) "Re-assess the computational cost using the same setting for all the models. Assessing the computational cost with flops would also help to improve the paper value."**
>
> In our initial manuscript, we reported the number of training samples and the GPUs used primarily to provide a basic understanding of the scale and resource requirements of our methods compared to existing methods. This information was aimed at giving readers a sense of the computational intensity relative to typical neural network solvers in this field.
>
> Following your suggestion, we have added additional efficiency metrics to Appendix A3.2.
> 1. FLOPs: We have now calculated the FLOPs for a forward pass of our models and the baselines, assuming the input data is 2D with 4 channels and resolution 128.
> 2. Inference time: Additionally, we have measured the average time required to predict a single step for a PDE instance during inference.
>
> We mainly compare with unified models with similar model sizes. The numbers can be found in the table below.
> |  | UPS-B (Ours)| MPP-B  | DPOT-M |
> |-------------------------------|-------------|--------|--------|
> | Num Params                    | 149M        | 116M   | 122M   |
> | Per Forward Pass FLOPs (G)    | 72.66       | 102.12 | 75.44  |
> | Single Step Inference Time (ms)| 1.77       | 2.34   | 1.88   |
>
> Compared to existing unified models with similar parameter counts, UPS has lower FLOPs and shorter inference time. This shows that our method is ideal for practical environments where both computational efficiency and speed are critical.

---

> ### Author Response · Authors · 2024-10-15
> **Updated Manuscript and Author Response (Cont.)**
>
> **(Major W3) "Reconsider using a more difficult dataset for resolution and zero/few-shot study to evaluate the model performance more appropriately."**
>
> Thank you for your constructive feedback. We acknowledge your concerns that the datasets in our initial experiments might not fully demonstrate our model’s capabilities in handling more complex and realistic scenarios. Based on your suggestion, we have conducted two additional sets of experiments.
> First, for few-shot generalization, we used the more challenging 2D compressible Navier-Stokes equations with $M=1$, $\eta = \zeta=0.1$* to better showcase our model's performance:
>
> |Num Adaptation Samples | 0|10  |100|Full|
> |:-:|:-:|:-:|:-:|:-:|
> |FNO|1.4302|0.567|0.3962|0.098|
> |ORCA|1.6399|0.1623|0.0898|0.0287|
> |UPS|0.0566|0.0134	|0.0039|0.0022|0.0009|
>
> These results show that UPS can generalize to challenging PDEs with limited training examples after being fine-tuned on PDE data from multiple families.
>
> Second, for zero-shot super-resolution, we added incompressible Navier-Stokes as well. Note that the PDEBench dataset has a maximum resolution of 512.
>
> | Test Resolution             | 256   | 512   |
> |-----------------------------|-------|-------|
> | Incomp Navier-Stokes (nRMSE)| 0.119 | 0.126 |
>
> These additional results and detailed analysis can be found in Table 2 and Table 3 of the revised manuscript.
>
> &nbsp;
>
> **(Minor W4) "Insufficient explanations."**
>
> Thank you for your valuable feedback. We have made the following revisions to the manuscript:
> - In Section 4, we added the pointer to the reference NLP dataset CoNLL-2003, which was chosen following the ORCA paper.
> - In Section 3.2, we clarified that the linear predictor is shared across all PDE families.
> - In our framework, data normalization is employed primarily to enhance model training stability and convergence. Normalizing the input data helps manage the numerical range that the neural network must handle, which is important given the wide variety of scales often present in PDE solutions. We acknowledge your correct observation that straightforward scaling transformations (like normalization by a constant factor) can alter the solutions of non-linear PDEs. To address this, after the neural network has processed the normalized data, we apply an inverse transformation to the outputs to revert them to the original scale. This step ensures that the final solutions presented and evaluated are consistent with the original PDEs. We consider experimenting with unnormalized inputs and using other techniques to prevent distortion of non-linear dynamics as future work.
> - In Section 3.1, we added the definition for the dimension $d^s$ and clarified its use in denoting coordinates.
> - In Section 5.2, we added "UPS-B refers to UPS with RoBERTa-Base" in the caption of Table 2.
>
> &nbsp;
>
> **(Minor W5) "Some description needs to be polished."**
>
> Thank you for your suggestions. We have made the following revisions to the paper:
> - In Section 3, we added the hat symbol $\hat u$ to the predictions to distinguish them from ground truths. We also double-checked the superscripts and subscripts of each  $u_t^s$.
> - As for Equation (1), we agree that the ideal way to write the definition of the operator is similar to the one that the reviewer mentioned. However, we follow the notational convention used in many previous works such as [1] and [2], and hence keep it consistent with the norms in ML-for-PDE literature.
> - Also for Equation (1), we use the definition of Dirichlet boundary condition with the value 0 to match with the most prevalent boundary conditions in the dataset. However, we agree that a general definition is more appropriate, so we have generalized the definition in the revised manuscript. Additionally, we have added a footnote for the reader to clarify that they can assume the value is 0 unless stated otherwise.
>
>
>
> &nbsp;
>
> **(Besides)**
> - Regarding the typo in the Shallow-Water Equations, we have added the missing terms to Appendix A2.7.
> - We removed the coloring from the caption of Figure 2.
>
> &nbsp;
>
> [1] PDE-Refiner: Achieving Accurate Long Rollouts with Neural PDE Solvers. Lippe et al. NeurIPS 2023.
>
> [2] DPOT: Auto-Regressive Denoising Operator Transformer for Large-Scale PDE Pre-Training. Hao et al. ICML 2024.

---

> > ### Comment · Reviewer_7gUR · 2024-10-16
> > **reply**
> >
> > Dear authors,
> >
> > Thank you for your hard effort on the manuscript. I am generally satisfied with the present version, with the exception of the following points, which are based on the
> > TMLR criterion "scientifically correct or not":
> >
> > 1. Because of the new robust computational cost assessment (Table 6 and 7), I believe Figure 1 is no longer necessary on the first page. I strongly recommend that the authors consider to replace Figure 1 to a visual representation (Figure) of either Table 6 or 7, or simply omitting Figure 1 entirely. As explained in my review, Figure 1 now seems to only confuse the reader and may lead to an unnecessary underevaluation of previous works.
> >
> > 2. I feel that my comment W4.3 has not yet been properly addressed. This is critical because it directly relates to the fundamental assumption of neural operators. In fact, the normalization process completely breaks this assumption completely. At the very least, I strongly urge the authors to discuss this issue in the manuscript more seriously to justify the work's correctness.
> >
> >
> > 3. Concerning the response to reviewer gB39 (manuscript: above Sec 5.1 in blue color), what is the intention of the authors on the following sentence?: $$\textbf{"but are averaged across test trajectories"}$$.
> > If this is simply mentioning that the test performance is averaged over the number of test samples (or trajectries), it seems too obvious and does not need explicit mention; Otherwise, the readers (myself included) may be confused about the authors' intended meaning.
> > If it implies something more than an average over test samples, I would appreciate a little more detailed explanation.

---

> > > ### Comment · Reviewer_7gUR · 2024-10-18
> > > **To the authors**
> > >
> > > Because review evaluation deadline has been arrived, I am going to wait for the authors' response by the end of today. I'll decide the evaluation based on the response: if the above points are appropriately addressed or not.
> > >
> > > Best,

---

> > > > ### Author Response · Authors · 2024-10-18
> > > > **Further revisions completed**
> > > >
> > > > Dear reviewer 7gUR,
> > > >
> > > > Thank you for your quick response and further suggestions! We have just uploaded a new version of our manuscript with the following changes:
> > > > 1. We removed Figure 1 from the main text.
> > > > 2. In Section 5 (page 6), we add a footnote to provide more detailing our normalization method. This footnote explains that normalization is necessary to maintain training stability when using pretrained LLM weights and acknowledges that it may affect non-linear equations. Our decision is based on the empirical observation that the scale of unstandardized PDE data can be quite large sometimes and can lead to the prediction and loss to explode, as we are loading pretrained weights for our models. Normalization has also been employed in existing works like MPP ([code](https://github.com/PolymathicAI/multiple_physics_pretraining/blob/462d2992b74563c0325d2ec8baf8cd8e7952b0bc/models/avit.py#L104)) and GNOT ([code](https://github.com/HaoZhongkai/GNOT/blob/5ee2e6925a43f9a340a6016bad4da2c82a452cbe/data_utils.py#L203)). We recognize the need for more theoretical justification and plan to investigate PDE-specific normalization techniques that preserve equation correctness in future work.
> > > > 3. The statement "are averaged across test trajectories" indeed means that the test performance is averaged over all test examples. We removed it from the paper to avoid confusion as suggested.
> > > >
> > > > We hope these revisions address all your concerns. Thank you once again for your invaluable input!

---

> ### Comment · Reviewer_7gUR · 2024-10-18
> **reply**
>
> I greatly appreciate the authors' last-minute efforts in addressing my comments. I am now satisfied with the quality of the paper and have provided an evaluation that reflects this.
>
> Best,

---

### Review · Reviewer_gB39 · 2024-09-18

**Summary Of Contributions:**

This paper presents Unified PDE Solvers (UPS), a network architecture
and state representation approach designed to permit one network to
implement neural operators for solving multiple PDEs. This
architecture incorporates a language model as part of an embedding to
encode the type of PDE and other parameters alongside a Fourier Neural
Operator to encode the current state. LLM-style transformer layers
process these to produce a final prediction for a future timestep. The
paper compares performance on several PDEBench tasks and against a
variety of baseline neural network operators.

**Audience:**

Yes

**Broader Impact Concerns:**

No concerns

**Claims And Evidence:**

No

**Requested Changes:**

**(Important)** The biggest question/concern I have is about the
contribution of the $h_{\text{meta}}$ from the LLM. I appreciate your
ablation study S3, but I'm still not sure whether the LLM is
contributing more than just providing a tag for the particular PDE.
Does the LLM's language pretraining help, or is its output just
helping label the PDE type for later stages of the network?

Did you try providing the parameters and PDE type in a different way
(i.e. a one-hot encoding per PDE or another opaque embedding)? Or
providing not the true PDE name, but instead some other "fake" name so
long as it's consistent? Is the specific PDE name and LLM adding some
additional capability that couldn't come from a different
embedding/encoding approach?

I'm just concerned that even with S3 it seems hard to rule that out.
Some of the results in table 4 for the cross-attention approach look
pretty close in performance to those with the LLM. I just wonder about
the impact of the language input.

**(Moderate)** I would be interested in information on any trends you
observed during fine-tuning (this is somewhat related to the first
note above). Broadly, how much of the final performance comes from the
task-specific fine tuning vs. the knowledge the LLM started with? I
appreciate that your experiments S1 and S2 look at this question. To
my eyes, the results in Table 4 seem to suggest that for certain
systems the task loss is very important (for example in the experiment
S2 Diff-Sorp 1D column, the largest drop appears to come with the task
loss), although it doesn't look like this trend is consistent across
tasks.

Even with the observed benefits of the language pretraining in S1 with
the fine-tuning it seems a bit unclear to me where where exactly the
improvements are coming from.

For the above information in Table 4 (and also for other result
tables), are the reported numbers averaged across several trials
(multiple, separately-trained networks)? I imagine they are averaged
across the test trajectories. It would help to have some sense of the
variance in some of these numbers since in some cases they look pretty
close.

**(Low)** When you quantify the computational costs, do you have some
idea of these beyond the number of training samples and number of
GPUs? What you report is a useful value to have, but something like a
FLOP count or time required for testing would also help compare the
costs of running these networks once they are trained, especially in
cases where your approach may be used many times after training.

**(Low)** In A.5.5 you mention having some issues with the pressure term
since other systems don't use these quantities and as a result the
model is trained to predict zeros. Would you expect this behavior
might get worse if this model were to be generalized to a wider set of
possible systems (so that more quantities might experience this bias)
and maybe benefit from some masking to indicate to the network the
relevant quantities to compute?

**Strengths And Weaknesses:**

**Strengths**

- Good variety of baseline networks
- Clear explanation of the approach and the different phases of evaluation and training
- Good effort to examine generalization capability and to conduct some ablation tests for different network components

**Weaknesses**

- The impact of the LLM embedding is still somewhat unclear, and I think would benefit from further testing
- I have some other concerns relating to some of the ablation tests
  (see below)
- Estimates of computational costs seem somewhat simple

---

> ### Author Response · Authors · 2024-10-15
> **Updated Manuscript and Author Response**
>
> Thank you for your review! Please find our response below.
>
> &nbsp;
>
> **(Important) "The biggest question/concern I have is about the contribution of the $h_{meta}$ from the LLM. I appreciate your ablation study S3, but I'm still not sure whether the LLM is contributing more than just providing a tag for the particular PDE. Does the LLM's language pretraining help, or is its output just helping label the PDE type for later stages of the network?"**
>
> Thank you for your question regarding the role of pretrained LLM in UPS. To investigate whether the LLM contributes beyond merely labeling the PDE type, we have added two ablation studies that use alternative embedding strategies which do not leverage language pretraining, according to your suggestions.
> 1. One-hot encodings: We replaced the LLM embeddings with one-hot encoded vectors representing each PDE type. This setting served as a baseline to assess the impact of merely labeling the PDE types without any semantic understanding.
> 2. Learnable random embeddings: We also tried embedding PDE families using a randomly initialized embedding layer that was trained from scratch along with the rest of the network, i.e., each new token of this additional embedding layer represents a PDE family.
> The results are shown below.
>
> ||1D Advection|1D Burgers |1D Diffusion-Sorption|1D Comp Navier-Stokes|2D Shallow-Water|2D Comp Navier-Stokes|2D Incomp Navier-Stokes|
> |:-:|:-:|:-:|:-:|:-:|:-:|:-:|:-:|
> |No Meta Data|0.0122|0.0453|0.001|0.0091|0.0026|0.0238|0.1171|
> |One-Hot Encodings|0.0029|0.0447|0.0011|0.006|0.0018|0.0198|0.095|
> |Learned-From-Scratch Embeddings|0.0041|0.0474|0.0014|0.0119|0.0036|0.0295|0.1103|
> |Pretrained Text Embeddings (UPS)| 0.0027 |0.0399 |0.0009 |0.0056 |0.0019 |0.0153 |0.0931|
>
> The model utilizing pretrained text embeddings consistently outperformed the one-hot and from-scratch embedding settings across PDE families. This suggests that the pretrained semantic knowledge in the LLM indeed contributes to processing the PDE data, not just in labeling PDE types but might also in understanding the underlying physics. However, the exact mechanism is still somewhat unclear, and we leave further investigation as future work. We also note that the one-hot setting outperforms the no-meta-data setting. This suggests that labeling the PDE families is still beneficial. However, the randomly initialized embeddings underperform the no-meta-data setting on a few datasets, such as Burgers and Diffusion-Sorption. We hypothesize that since the embedding layer is learned from scratch, it might initially add noise to the training process.  In terms of learning dynamics, we also observed that using text embeddings demonstrated faster convergence compared to the alternative strategies.
>
> We appreciate this opportunity to further validate our model's design and have included these additional results in Table 4 S3 of the revised manuscript.
>
> &nbsp;
>
> **(Moderate) "I would be interested in information on any trends you observed during fine-tuning (this is somewhat related to the first note above). Broadly, how much of the final performance comes from the task-specific fine tuning vs. the knowledge the LLM started with?"**
>
> Thank you for your question regarding the relative contributions of task-specific fine-tuning vs. the pretrained knowledge of LLMs to the performance of UPS. We believe that task-specific fine-tuning plays a more important role in UPS because, without fine-tuning, the pretrained LLM simply cannot model any PDEs due to the large domain gap.  It is the data-driven fine-tuning that effectively repurposes LLMs into PDE solvers. The pretrained knowledge of the LLM acts more like an "add-on" to the fine-tuning process, enabling better final performance and faster convergence. This is supported by Experiment S1 in Table 4, where the fine-tuned model outperformed the train-from-scratch setting. However, the fact that the train-from-scratch baseline also performs reasonably shows that while the pretrained knowledge of LLM is beneficial, it is not as important as fine-tuning.

---

> ### Author Response · Authors · 2024-10-15
> **Updated Manuscript and Author Response (Cont.)**
>
> **(Moderate) "For the above information in Table 4 (and also for other result tables), are the reported numbers averaged across several trials (multiple, separately-trained networks)? I imagine they are averaged across the test trajectories."**
>
> We acknowledge that ideally, results should be averaged over multiple runs to account for training variability. However, due to computational constraints as an academic lab and the large datasets involved, each result reported in our study is based on a single run per network configuration. Nonetheless, results are indeed averaged across test trajectories. We clarified this in the beginning of Section 5 in the revised paper.
>
> Despite the limitation of single-run results, we have taken steps to ensure the reliability and reproducibility of our findings. For instance, for the out-of-domain generalization experiments in Table 2, we selected multiple PDE datasets to assess the stability and generalizability of our model's performance. As part of our future work, we aim to use additional resources to perform a more comprehensive evaluation, including multiple runs to better capture the variability and confirm the robustness of our results.
>
> &nbsp;
>
> **(Low) "When you quantify the computational costs, do you have some idea of these beyond the number of training samples and number of GPUs? What you report is a useful value to have, but something like a FLOP count or time required for testing would also help compare the costs of running these networks once they are trained, especially in cases where your approach may be used many times after training."**
>
> In our initial manuscript, we reported the number of training samples and the GPUs used primarily to provide a basic understanding of the scale and resource requirements of our methods compared to existing methods. This information was aimed at giving readers a sense of the computational intensity relative to typical neural network solvers in this field.
>
> Following your suggestion, we have added additional efficiency metrics to Appendix A3.2.
> 1. FLOPs: We have now calculated the FLOPs for a forward pass of our models and the baselines, assuming the input data is 2D with 4 channels and resolution 128.
> 2. Inference time: Additionally, we have measured the average time required to predict a single step for a PDE instance during inference.
>
> We mainly compare with unified models with similar model sizes. The numbers can be found in the table below.
> |  | UPS-B (Ours)| MPP-B  | DPOT-M |
> |-------------------------------|-------------|--------|--------|
> | Num Params                    | 149M        | 116M   | 122M   |
> | Per Forward Pass FLOPs (G)    | 72.66       | 102.12 | 75.44  |
> | Single Step Inference Time (ms)| 1.77       | 2.34   | 1.88   |
>
> Compared to existing unified models with similar parameter counts, UPS has lower FLOPs and shorter inference time. This shows that our method is ideal for practical environments where both computational efficiency and speed are critical.
>
> &nbsp;
>
> **(Low) "In A.5.5 you mention having some issues with the pressure term since other systems don't use these quantities and as a result the model is trained to predict zeros. Would you expect this behavior might get worse if this model were to be generalized to a wider set of possible systems (so that more quantities might experience this bias) and maybe benefit from some masking to indicate to the network the relevant quantities to compute?"**
>
> Good question! We believe that this issue you mentioned is related to the PDE families represented in the training data. For UPS to work well, it is crucial to ensure sufficient representation of each physical quantity that our model is designed to handle. For instance, consider the pressure term discussed in our paper. We believe that adding more PDEs with pressure terms will help alleviate this issue, and finding more families with pressure terms is feasible since pressure is used to model a lot of general PDEs like fluid-dynamics and wave equations under different mediums. Thus, enriching our training data with more examples from different families will make the observed issue less pronounced.

---

> > ### Comment · Reviewer_gB39 · 2024-10-21
> >
> > Thank you for your thorough responses to my questions and for all your
> > work on these revisions. I really appreciate the effort involved in
> > quickly running the additional ablation tests. I think they have
> > helped put the results of your paper into a clearer context so thank
> > you again for their inclusion. I also appreciate your clarification on
> > the averaging in the tests and the included FLOP and evaluation time
> > information. I absolutely understand computational constraints
> > limiting your ability to run additional full training and test passes,
> > but I also appreciate making what's averaged in the results clearer.
> >
> > I definitely agree with your assessment that there are still a lot of
> > questions that could examined in future work and I hope some
> > additional follow-up evaluation there will be possible.
> >
> > One further addition that I think would be welcome (if you have the
> > data available) is a variance or standard deviation for some of the
> > result tables (in particular for tables 2 and 4). If you have this
> > available and could include it in an appendix section, I think that
> > would help compare how the distributions between the trajectories
> > change in some of these experiments. But this is a quite minor request
> > and certainly not critical.

---

### Author Response · Authors · 2024-09-23
**Update on planned timeline for making the requested revisions**

We are extremely grateful for the constructive feedback from the reviewers and are planning to run additional experiments to address shared concerns regarding (1) better understanding the language component and the effect of PDE metadata in fine-tuning; (2) improving the rigorousness of evaluation by adding autoregressive evaluation and running multiple sets of out-of-domain generalization experiments; and (3) better quantifying the computational costs and performing a fair comparison with the baselines. However, these additional results require a decent amount of compute. Since our lab is using a shared cluster at our department and the cluster is almost unavailable in the coming two weeks due to the ICLR deadline, we are planning to run the experiments right after the ICLR deadline (Oct 1st). We will make sure to update the results with the reviewers as soon as possible and no later than the 4-week deadline of the TMLR review process. Thank you very much!

---

### Author Response · Authors · 2024-10-15
**Updated Manuscript and Supplementary Material**

We are grateful for the constructive reviews. We have revised the manuscript in response to feedback from reviewers. Key updates include:

1. **More Ablation Studies:** We present additional ablation studies in Table 4 S3 to assess the contribution of pretrained LLMs beyond merely labeling PDE types. These studies utilized one-hot encodings and learnable random embeddings as alternative strategies, demonstrating the benefits of using pretrained text embeddings when processing PDE data.
2. **More Generalizability Results:** We added experiments using more challenging datasets, such as the 2D Navier-Stokes equations, to Table 2 and Table 3 of the paper. These experiments aimed to showcase the model's ability to generalize to unseen PDEs and provided evidence of its few-shot learning capabilities.
3. **Computational Efficiency Metrics:** Additional efficiency metrics, including FLOPs and inference times, were reported in Appendix A3.2 to better quantify the efficiency of our method.
4. **Clarifications and Corrections:** Various sections of the manuscript were revised for clarity and correctness. Notable changes include correcting notation inconsistencies and refining the description of the model’s evaluation methods.

We have highlighted the revised text in blue for clarity. Additionally, we have included the autoregressive evaluation script in the supplementary material. We have also provided separate responses to each reviewer's comments.

---

### Decision · Action_Editor_cymt · 2024-10-23

**Recommendation:** Accept as is

**Comment:**

The paper proposes an interesting, and even ambitious, cross-domain study of combining LLMs with PDE solvers. The reviews and discussion was insightful and supports acceptance.

**Audience:**

Yes

**Claims And Evidence:**

Yes